# Efficient Attention via Control Variates

**Lin Zheng**[1]* **Jianbo Yuan**[2] **Chong Wang**[3]* **Lingpeng Kong**[1]
[1]The University of Hong Kong  [2]ByteDance Inc.  [3]Apple Inc.
{lzheng2,lpk}@cs.hku.hk
jianbo.yuan@bytedance.com  mr.chongwang@apple.com

## Abstract

Random-feature-based attention (RFA) is an efficient approximation of softmax attention with linear runtime and space complexity. However, the approximation gap between RFA and conventional softmax attention is not well studied. Built upon previous progress of RFA, we characterize this gap through the lens of control variates and show that RFA can be decomposed into a sum of multiple control variate estimators for each element in the sequence. This new framework reveals that exact softmax attention can be recovered from RFA by manipulating each control variate. Besides, it allows us to develop a more flexible form of control variates, resulting in a novel attention mechanism that significantly reduces the approximation gap while maintaining linear complexity. Extensive experiments demonstrate that our model outperforms state-of-the-art efficient attention mechanisms on both vision and language tasks.[1]

## 1 Introduction

Random-feature-based attention (RFA, also known as Performer; Choromanski et al., 2021; Peng et al., 2021b) is an established fast approximation to the conventional softmax attention mechanism (Bahdanau et al., 2014; Vaswani et al., 2017), which successfully scales Transformer models to processing much longer sequences (Choromanski et al., 2021). At its core is the usage of random features (RF; Rahimi & Recht, 2008) to linearize the exponential kernel in softmax attention, which reduces the computational cost from quadratic to linear runtime and space complexity. Despite its efficiency, recent studies have pointed out that such approximation suffers from substantial performance degeneration (Xiong et al., 2021a; Zheng et al., 2022b).

In this work, we generalize the formulation of RFA via control variates (Owen, 2013), which characterizes the approximation gap between RFA and softmax attention in theory. We first show that RFA can be decomposed from a *global* approximation over the whole sequence into a sum of *local* control variate estimators, each of which is applied to an individual element in the sequence. Under this formulation, RFA is equivalent to employing the same coefficient for all control variate estimators to scale their variance isotropically (§3.1). Besides, we prove that if we optimize the coefficient of each control variate to minimize the estimation variance individually, RFA estimation becomes exact, that is, softmax attention is recovered with zero bias and zero variance (§3.2).

Our key observation is that such formulation reveals a localized perspective of the RFA approximation. Instead of directly seeking a better estimate over the entire sequence, we can break down the problem into smaller problems that aim at improving the approximation for each subsequence (§4). The control variate estimator for each subsequence can be tuned separately and combined to yield better estimation, which provably reduces approximation error in the global sense (§4.1). Nevertheless, one caveat is that as the number of sub-problems increases, the approximation gap will be reduced but at the expense of higher computational complexity. For instance, if we optimize the control variate for every single element, softmax attention would be recovered as desired but with quadratic complexity. To attain a good trade-off between approximation quality and efficiency, we develop a new **E**fficient attention via control **VA**riates (**EVA**) that implements this divide-and-conquer strategy efficiently. In EVA, the sequence is partitioned into a fixed number of disjoint subsets. For the subset

---

*The majority of this work was done while these authors were at Bytedance.

[1]Our code and models are available at this link.

that might bear the highest correlations to the query, we explicitly optimize the control variate for each element, which recovers exact softmax attention probabilities; while for the others, the control variate coefficient is shared locally among all elements within the same subset. The resulting attention mechanism is not only highly effective but also runs with the same computational complexity as RFA (§4.2). Extensive experiments on both language and vision tasks demonstrate that EVA outperforms the state-of-the-art efficient attention methods (§5).

## 2 BACKGROUND

### 2.1 SOFTMAX ATTENTION MECHANISM

Assume there exist a set of $N$ queries $\{\mathbf{q}_n\}_{n=1}^N$ and $M$ key-value pairs $\mathbf{K} = [\mathbf{k}_1, \ldots, \mathbf{k}_M]$ and $\mathbf{V} = [\mathbf{v}_1, \ldots, \mathbf{v}_M]$, where queries, keys and values are all $d$-dimensional vectors. The softmax attention mechanism (Bahdanau et al., 2014; Vaswani et al., 2017) is defined as an average over the value vectors weighted by the dot-product similarities of the queries and keys. For the $n$-th query, the attention mechanism outputs

$$\text{SoftmaxAttn}(\mathbf{q}_n, \mathbf{K}, \mathbf{V}) := \sum_{m=1}^M \frac{\exp\left(\mathbf{q}_n^\top \mathbf{k}_m\right)}{\sum_{m'=1}^M \exp\left(\mathbf{q}_n^\top \mathbf{k}_{m'}\right)} \mathbf{v}_m. \tag{1}$$

In the case of self-attention (Lin et al., 2017; Vaswani et al., 2017), we have $M = N$, which results in quadratic computational complexity since we have to compute the similarity for each query-key pair explicitly.

### 2.2 RANDOM-FEATURE-BASED ATTENTION WITH SELF-NORMALIZED IMPORTANCE SAMPLING

Recently, Zheng et al. (2022b) identifies that softmax attention (Equation 1) can be written as an expectation over an attention-like aggregating function,

$$\text{SoftmaxAttn}(\mathbf{q}_n, \mathbf{K}, \mathbf{V}) = \sum_{m=1}^M \frac{\exp\left(\mathbf{q}_n^\top \mathbf{k}_m\right)}{\sum_{m'=1}^M \exp\left(\mathbf{q}_n^\top \mathbf{k}_{m'}\right)} \mathbf{v}_m = \mathbb{E}_{\omega \sim p_n(\omega)}\left[f_n(\omega)\right], \tag{2}$$

where

$$f_n(\omega) := \frac{\sum_{m=1}^M \xi(\mathbf{q}_n, \omega)\xi(\mathbf{k}_m, \omega)\mathbf{v}_m}{\sum_{m'=1}^M \xi(\mathbf{q}_n, \omega)\xi(\mathbf{k}_{m'}, \omega)}, \quad p_n(\omega) := \frac{\mathcal{N}(\omega; 0, \mathbf{I}) \sum_{m=1}^M \xi(\mathbf{q}_n, \omega)^\top \xi(\mathbf{k}_m, \omega)}{Z}. \tag{3}$$

Here $\xi(\cdot, \cdot)$ is the *randomized mapping* defined in such a way that $\exp\left(\mathbf{q}_n^\top \mathbf{k}_m\right) = \mathbb{E}_{\omega \sim \mathcal{N}(0, \mathbf{I})}\left[\xi(\mathbf{q}_n, \omega)^\top \xi(\mathbf{k}_m, \omega)\right]$, and $Z = \sum_{m=1}^M \exp\left(\mathbf{q}_n^\top \mathbf{k}_m\right)$ denotes the normalizing constant of distribution $p_n$. Throughout this paper, we consider the positive randomized mapping $\xi(\mathbf{x}, \omega) = \exp\left(\omega^\top \mathbf{x} - \frac{1}{2}\|\mathbf{x}\|^2\right)$ (Choromanski et al., 2021) unless otherwise specified.

Random-Feature-based Attention (RFA) methods (Choromanski et al., 2021; Peng et al., 2021b) can be interpreted as performing *self-normalized importance sampling* (SNIS; Hesterberg, 1995) to approximate Equation 2 (Zheng et al., 2022b). In SNIS, one draws Monte Carlo samples from some *proposal* distribution $q(\omega)$ instead of the true distribution $p_n(\omega)$ and estimates the target expectation as $\mathbb{E}_{\omega \sim p_n(\omega)}\left[f_n(\omega)\right] = \mathbb{E}_{\omega \sim q(\omega)}\left[\frac{p_n(\omega)}{q(\omega)} f_n(\omega)\right] \approx \frac{\sum_{s=1}^S \frac{p_n(\omega)}{q(\omega)} f_n(\omega_s)}{\sum_{s=1}^S \frac{p_n(\omega_s)}{q(\omega_s)}}$, where $\omega_1, \ldots, \omega_S \sim q(\omega)$. Vanilla RFA amounts to constructing the SNIS estimation with $q(\omega) = \mathcal{N}(\omega; 0, \mathbf{I})$. The SNIS representation also turns out equivalent to the more established form of RFA,

$$\text{RFA}(\mathbf{q}_n, \mathbf{K}, \mathbf{V}) := \frac{\sum_{s=1}^S \frac{p_n(\omega_s)}{q(\omega_s)} f(\omega_s)}{\sum_{s=1}^S \frac{p_n(\omega_s)}{q(\omega_s)}} = \frac{\sum_{m=1}^M \phi(\mathbf{q}_n, \omega)^\top \phi(\mathbf{k}_m, \omega)\mathbf{v}_m}{\sum_{m'=1}^M \phi(\mathbf{q}_n, \omega)^\top \phi(\mathbf{k}_{m'}, \omega)}, \tag{4}$$

where the *random feature*, denoted by $\phi(\mathbf{x}, \omega) := 1/\sqrt{S}[\xi(\mathbf{x}, \omega_1), \ldots, \xi(\mathbf{x}, \omega_S)]^\top$, is proposed to approximate exponential kernels in its original motivation (see Appendix A for a detailed review).

## 2.3 CONTROL VARIATES

Control variates aim to reduce the estimation variance of an expectation $\mathbb{E}\left[g(\boldsymbol{\omega})\right]$. Assuming our original RFA estimation is $g(\boldsymbol{\omega}) \in \mathbb{R}^d$ and there is some *control variate* $h(\boldsymbol{\omega}) \in \mathbb{R}$ with a known expectation $\mathbb{E}\left[h(\boldsymbol{\omega})\right]$, we can employ the control variate $h(\boldsymbol{\omega})$ with the coefficient $\boldsymbol{\beta} \in \mathbb{R}^d$ as follows,

$$\widetilde{g}(\boldsymbol{\omega}) = g(\boldsymbol{\omega}) - \boldsymbol{\beta} h(\boldsymbol{\omega}) + \boldsymbol{\beta}\mathbb{E}\left[h(\boldsymbol{\omega})\right] \tag{5}$$

Note that the resulting estimator remains unbiased since $\mathbb{E}\left[\widetilde{g}(\boldsymbol{\omega})\right] = \mathbb{E}\left[g(\boldsymbol{\omega})\right] - \boldsymbol{\beta}\mathbb{E}\left[h(\boldsymbol{\omega})\right] + \boldsymbol{\beta}\mathbb{E}\left[h(\boldsymbol{\omega})\right] = \mathbb{E}\left[g(\boldsymbol{\omega})\right]$. However, the estimation variance can be largely reduced if $g(\cdot)$ and the scaled control variate $\boldsymbol{\beta} h(\boldsymbol{\omega})$ are positively correlated (Owen, 2013).

## 3 DISSECTING RFA WITH CONTROL VARIATES

In this section, we first go through the connections among RFA, importance sampling, and control variates, revealing a decomposed formulation of RFA (§3.1), and then quantify the approximation gap between RFA and softmax attention (§3.2) from these connections.

## 3.1 RFA AS A SUM OF LOCAL CONTROL VARIATE ESTIMATORS

As shown in Equation 4, RFA estimation considers all key-value pairs and produces a *global* approximation over the entire sequence. In contrast, our work develops a *decomposed* representation of RFA based on the recent advances in SNIS (Vlassis et al., 2021), which indicates that an SNIS estimate is asymptotically equivalent to a control variate estimate (the detailed derivations is deferred to Appendix B.2). In particular, we have

$$\frac{\sum_{s=1}^{S} \frac{p_n(\omega_s)}{q(\omega_s)} f(\omega_s)}{\sum_{s=1}^{S} \frac{p_n(\omega_s)}{q(\omega_s)}} = \frac{1}{S}\sum_{s=1}^{S} \frac{p_n(\omega_s)}{q(\omega_s)} f(\omega_s) - \frac{\sum_{s=1}^{S} \frac{p_n(\omega_s)}{q(\omega_s)} f(\omega_s)}{\sum_{s=1}^{S} \frac{p_n(\omega_s)}{q(\omega_s)}} \left(\frac{1}{S}\sum_{s=1}^{S} \frac{p_n(\omega_s)}{q(\omega_s)} - 1\right)$$

$$:= g(\boldsymbol{\omega}) - \widehat{\boldsymbol{\beta}}(\boldsymbol{\omega})\left(h(\boldsymbol{\omega}) - \mathbb{E}\left[h(\boldsymbol{\omega})\right]\right) := \widetilde{g}(\boldsymbol{\omega}), \tag{6}$$

where $g(\boldsymbol{\omega}) := \frac{1}{S}\sum_{s=1}^{S} \frac{p_n(\omega_s)}{q(\omega_s)} f(\omega_s)$ is our base estimate, $h(\boldsymbol{\omega}) := \frac{1}{S}\sum_{s=1}^{S} \frac{p_n(\omega_s)}{q(\omega_s)}$ is the control variate with control coefficient $\widehat{\boldsymbol{\beta}}(\boldsymbol{\omega}) := \left(\sum_{s=1}^{S} \frac{p_n(\omega_s)}{q(\omega_s)} f(\omega_s)\right) \Big/ \left(\sum_{s=1}^{S} \frac{p_n(\omega_s)}{q(\omega_s)}\right) = \frac{g(\boldsymbol{\omega})}{h(\boldsymbol{\omega})}$.

We now examine the formulation of $g(\cdot)$ and $h(\cdot)$ in the context of RFA. According to Equation 3,

$$g(\boldsymbol{\omega}) = \frac{1}{S}\sum_{s=1}^{S} \frac{p_n(\omega_s)}{q(\omega_s)} f(\omega_s) = \sum_{s=1}^{S} \alpha(\omega_s) \sum_{m=1}^{M} \xi(\mathbf{q}_n, \omega_s)\xi(\mathbf{k}_m, \omega_s)\mathbf{v}_m,$$

$$h(\boldsymbol{\omega}) = \frac{1}{S}\sum_{s=1}^{S} \frac{p_n(\omega_s)}{q(\omega_s)} = \sum_{s=1}^{S} \alpha(\omega_s) \sum_{m=1}^{M} \xi(\mathbf{q}_n, \omega_s)\xi(\mathbf{k}_m, \omega_s),$$

where $\alpha(\omega_s) := \frac{1}{S}\frac{\mathcal{N}(\omega_s; 0, \mathbf{I})}{Z q(\omega_s)}$ collects terms that is constant w.r.t. queries, keys, and values. Our key observation is that by changing the order of summations, both $g(\cdot)$ and $h(\cdot)$ can be decomposed as $g(\boldsymbol{\omega}) = \sum_{m=1}^{M} g_m(\boldsymbol{\omega})$ and $h(\boldsymbol{\omega}) = \sum_{m=1}^{M} h_m(\boldsymbol{\omega})$ respectively, where

$$g_m(\boldsymbol{\omega}) = \sum_{s=1}^{S} \alpha(\omega_s)\xi(\mathbf{q}_n, \omega_s)\xi(\mathbf{k}_m, \omega_s)\mathbf{v}_m, \qquad h_m(\boldsymbol{\omega}) = \sum_{s=1}^{S} \alpha(\omega_s)\xi(\mathbf{q}_n, \omega_s)\xi(\mathbf{k}_m, \omega_s).$$

As a result, we can decompose the entire RFA estimate in Equation 6 into a summation of $M$ control variate estimates following

$$\widetilde{g}(\boldsymbol{\omega}) = g(\boldsymbol{\omega}) - \widehat{\boldsymbol{\beta}}(\boldsymbol{\omega})\left(h(\boldsymbol{\omega}) - \mathbb{E}\left[h(\boldsymbol{\omega})\right]\right)$$

$$= \left(\sum_{m=1}^{M} g_m(\boldsymbol{\omega})\right) - \widehat{\boldsymbol{\beta}}(\boldsymbol{\omega})\left(\left(\sum_{m=1}^{M} h_m(\boldsymbol{\omega})\right) - \mathbb{E}\left[\sum_{m=1}^{M} h_m(\boldsymbol{\omega})\right]\right)$$

$$= \sum_{m=1}^{M} g_m(\boldsymbol{\omega}) - \widehat{\boldsymbol{\beta}}(\boldsymbol{\omega})\left(h_m(\boldsymbol{\omega}) - \mathbb{E}\left[h_m(\boldsymbol{\omega})\right]\right) := \sum_{m=1}^{M} \widetilde{g}_m(\boldsymbol{\omega}). \tag{7}$$

Here $\widetilde{g}_m(\boldsymbol{\omega}) = g_m(\boldsymbol{\omega}) - \widehat{\boldsymbol{\beta}}(\boldsymbol{\omega})\left(h_m(\boldsymbol{\omega}) - \mathbb{E}\left[h_m(\boldsymbol{\omega})\right]\right)$ denotes the corresponding control variate estimator of the $m$-th key-value pair,[2] and $\widehat{\boldsymbol{\beta}}(\boldsymbol{\omega})$ is the coefficient shared across the entire sequence.

## 3.2 Optimizing Coefficients in RFA Locally Recovers Softmax Attention

Based on the decomposition of RFA in Equation 7, we have one local control variate attached to each key-value pair. To see the benefit of such decomposition, we demonstrate that softmax attention is equivalent to associating each control variate with a locally optimized coefficient $\widehat{\boldsymbol{\beta}}_m$ in RFA.

**Proposition 1.** *Let $\widetilde{g}_m(\boldsymbol{\omega}) = g_m(\boldsymbol{\omega}) - \widehat{\boldsymbol{\beta}}_m\left(h_m(\boldsymbol{\omega}) - \mathbb{E}\left[h_m(\boldsymbol{\omega})\right]\right)$. We denote the variance of some estimator $g(\boldsymbol{\omega})$ as $\mathrm{Var}\left[g(\boldsymbol{\omega})\right] := \mathrm{Cov}\left[g(\boldsymbol{\omega}), g(\boldsymbol{\omega})\right]$. Then the optimal $\widehat{\boldsymbol{\beta}}_m$ that minimizes $\mathrm{Tr}\left(\mathrm{Var}\left[\widetilde{g}_m(\boldsymbol{\omega})\right]\right)$ (i.e., the sum variance over all dimensions) is of the form*

$$\boldsymbol{\beta}_m^* := \arg\min_{\boldsymbol{\beta}} \mathrm{Tr}\left(\mathrm{Var}\left[\widetilde{g}_m(\boldsymbol{\omega})\right]\right) = \mathbf{v}_m = \frac{g_m(\boldsymbol{\omega})}{h_m(\boldsymbol{\omega})}. \tag{8}$$

*Furthermore, by letting $\widehat{\boldsymbol{\beta}}_m = \boldsymbol{\beta}_m^*$ for all $m = 1, 2, \ldots, M$, we have $\mathrm{Tr}\left(\mathrm{Var}\left[\widetilde{g}_m(\boldsymbol{\omega})\right]\right) = 0$. As a result, $\mathrm{Tr}\left(\mathrm{Var}\left[\widetilde{g}(\boldsymbol{\omega})\right]\right) = 0$ and thus $\mathsf{RFA}(\mathbf{q}_n, \mathbf{K}, \mathbf{V}) = \widetilde{g}(\boldsymbol{\omega}) = \mathsf{SoftmaxAttn}(\mathbf{q}_n, \mathbf{K}, \mathbf{V})$.*

The proof is deferred to Appendix B.4. This proposition implies optimizing $\widehat{\boldsymbol{\beta}}_m$ for each key-value pair in the decomposed formulation of RFA recovers the exact softmax attention. It not only characterizes the theoretical gap introduced by RFA but also sheds light on how to improve RFA towards softmax attention from a localized perspective. Furthermore, it delineates the trade-off between estimation quality and computational costs. On the one hand, if we use a distinct $\widehat{\boldsymbol{\beta}}_m$ for each estimator, we could achieve a perfect estimation, albeit at the expense of computing $\exp \mathbf{q}_n^\top \mathbf{k}_m$ for every query-key pair explicitly with quadratic time and space complexity. On the other hand, if a single shared coefficient is employed, it degrades to conventional RFA, where all the control variate estimators can be merged and computed together in linear complexity (Choromanski et al., 2021; Peng et al., 2021b; Zheng et al., 2022b).

## 4 EVA: Efficient Attention via Control Variates

In this section, we demonstrate that the control variate formulation offers a natural way to improve RFA with a finer-grained treatment over control variates. We describe the improved efficient attention mechanism EVA in §4.1 and its practical implementation in §4.2.

## 4.1 Control Variates with Locally Shared Coefficients

We denote $[M] := \{1, 2, \ldots, M\}$ as the set of all key-value indices. Instead of employing the same coefficient for all control variates as in RFA, we propose to partition $[M]$ into $C$ subsets $\mathcal{P}_1, \mathcal{P}_2, \ldots, \mathcal{P}_C$ and allocate a *locally* shared $\boldsymbol{\beta}_c$ for each subset $\mathcal{P}_c$. For all $\boldsymbol{\beta}_c$ and their optimum $\boldsymbol{\beta}_m^*$ for each token, define the weighted mean squared error (weighted MSE) as $\sum_{c=1}^{C} \sum_{m \in \mathcal{P}_c} \alpha_m \|\boldsymbol{\beta}_c - \boldsymbol{\beta}_m^*\|^2$, where $\alpha_m > 0$ and $\sum_{c=1}^{C} \sum_{m \in \mathcal{P}_c} \alpha_m = 1$. To see the benefit of partitioning, we demonstrate that there always exists some $\{\boldsymbol{\beta}_c\}_{c=1}^{C}$ that achieves lower weighted MSE than any globally shared coefficient (see Appendix B.5 for a formal argument). The next question is how to determine $\{\boldsymbol{\beta}_c\}_{c=1}^{C}$. According to Proposition 1, a natural choice is to adapt the optimal coefficients (Equation 8) to the case of partitioned subsets. We justify this choice by proving that it is also optimal in minimizing the MSE above weighted by the true attention probabilities.

**Proposition 2.** *Suppose $U$ is a set of key-value indices, $\boldsymbol{\beta}_m^*$ is the optimal coefficient for each $m \in U$ as defined in Proposition 1, and $\mathcal{P}_1, \mathcal{P}_2, \ldots, \mathcal{P}_C$ are an arbitrary partition of $U$, where each subset $\mathcal{P}_c$ is associated with a distinct $\boldsymbol{\beta}_c$. We consider the following weighted mean squared error,*

$$J(\boldsymbol{\beta}_1, \ldots, \boldsymbol{\beta}_C) := \sum_{c=1}^{C} \sum_{m \in \mathcal{P}_c} \frac{\exp\left(\mathbf{q}_n^\top \mathbf{k}_m\right)}{\sum_{m' \in U} \exp\left(\mathbf{q}_n^\top \mathbf{k}_{m'}\right)} \|\boldsymbol{\beta}_c - \boldsymbol{\beta}_m^*\|^2. \tag{9}$$

---

[2]Note that the expectation of individual control variates $h_m(\cdot)$ is still in closed form as $\mathbb{E}\left[h_m(\boldsymbol{\omega})\right] = \exp(\mathbf{q}_n^\top \mathbf{k}_m)/Z$. The derivation can be found in Appendix B.3.

*Then for each $c = 1, \ldots, C$ we have*

$$\boldsymbol{\beta}_c^* := \arg\min_{\boldsymbol{\beta}_c} J(\boldsymbol{\beta}_1, \ldots, \boldsymbol{\beta}_C) = \frac{\mathbb{E}\left[\sum_{m \in \mathcal{P}_c} g_m(\boldsymbol{\omega})\right]}{\mathbb{E}\left[\sum_{m \in \mathcal{P}_c} h_m(\boldsymbol{\omega})\right]}. \tag{10}$$

*As a consequence, with $\boldsymbol{\beta}_c = \boldsymbol{\beta}_c^*$, the partition scheme must achieve lower weighted mean squared error than any globally shared $\boldsymbol{\beta}$, that is, $J(\boldsymbol{\beta}_1 = \boldsymbol{\beta}_1^*, \ldots, \boldsymbol{\beta}_C = \boldsymbol{\beta}_C^*) \leq J(\boldsymbol{\beta}_1 = \boldsymbol{\beta}, \ldots, \boldsymbol{\beta}_C = \boldsymbol{\beta})$.*

The proof can be found in Appendix B.6. Apart from measuring the squared errors for all coefficients, Equation 9 also governs the significance of each error by its corresponding softmax weights, which attains closer alignment with true softmax attention. Therefore, this proposition implies that it is much easier for the partitioned control variate estimators to obtain coefficients closer to their optimum while faithfully respecting softmax attention. The optimal coefficients $\boldsymbol{\beta}_c^*$ could be estimated via Monte Carlo samples as $\boldsymbol{\beta}_c^* \approx \widehat{\boldsymbol{\beta}}_c(\boldsymbol{\omega}) = \left(\sum_{m \in \mathcal{P}_c} g_m(\boldsymbol{\omega})\right) / \left(\sum_{m \in \mathcal{P}_c} h_m(\boldsymbol{\omega})\right)$, which is a widely adopted strategy in the control variate literature (Wang et al., 2013; Owen, 2013). The resulting estimator for each subset $\mathcal{P}_c$ takes the form

$$\sum_{m \in \mathcal{P}_c} \left( g_m(\boldsymbol{\omega}) - \widehat{\boldsymbol{\beta}}_c(\boldsymbol{\omega}) h_m(\boldsymbol{\omega}) + \widehat{\boldsymbol{\beta}}_c(\boldsymbol{\omega}) \frac{\exp(\mathbf{q}_n^\top \mathbf{k}_m)}{Z} \right) = \frac{\sum_{m \in \mathcal{P}_c} \exp(\mathbf{q}_n^\top \mathbf{k}_m)}{Z} \widehat{\boldsymbol{\beta}}_c(\boldsymbol{\omega}). \tag{11}$$

**Partially Optimized Coefficients.** Given the optimality of using a separate coefficient for each key-value pair, we could further improve the estimation by selecting some subset $E \subseteq [M]$ and employ $\widehat{\boldsymbol{\beta}}_m = \widehat{\boldsymbol{\beta}}_m^* = \mathbf{v}_m$ for each $m \in E$. Without loss of generality, we assume $E \cap \mathcal{P}_c = \varnothing$ for all $c = 1, \ldots, C$ and $[M] = \left(\bigcup_{c=1}^C \mathcal{P}_c\right) \cup E$. According to Proposition 1, for each $m \in E$ we have

$$\widetilde{g}_m(\boldsymbol{\omega}) = g_m(\boldsymbol{\omega}) - \widehat{\boldsymbol{\beta}}_m h_m(\boldsymbol{\omega}) + \widehat{\boldsymbol{\beta}}_m \frac{\exp(\mathbf{q}_n^\top \mathbf{k}_m)}{Z} = \frac{\exp(\mathbf{q}_n^\top \mathbf{k}_m) \mathbf{v}_m}{Z}. \tag{12}$$

We choose $E$ by running an additional sparse attention mechanism (e.g., local window attention (Child et al., 2019) or Reformer (Kitaev et al., 2020)), which tend to select tokens that are more relevant to the query in sub-quadratic complexity. Since estimates on these critical tokens are exact, this strategy not only reduces the overall squared error (Equation 9), but also produces a more informative context for queries, which often translates into better empirical performance. Combining Equations 12 and 11 together, we obtain an improved **E**fficient attention via control **VA**riates (**EVA**),

$$\text{EVA}(\mathbf{q}_n, \mathbf{K}, \mathbf{V}) := \widetilde{g}(\boldsymbol{\omega}) = \sum_{m \in E} \widetilde{g}_m(\boldsymbol{\omega}) + \sum_{m \notin E} \widetilde{g}_m(\boldsymbol{\omega})$$

$$= \sum_{m \in E} \frac{\exp(\mathbf{q}_n^\top \mathbf{k}_m)}{Z} \mathbf{v}_m + \sum_{c=1}^C \frac{\sum_{m \in \mathcal{P}_c} \exp(\mathbf{q}_n^\top \mathbf{k}_m)}{Z} \widehat{\boldsymbol{\beta}}_c(\boldsymbol{\omega}). \tag{13}$$

**Comparison with Vanilla RFA.** EVA and vanilla RFA can be re-written in a similar way (see Appendix B.7 for a detailed derivation),

$$\text{RFA}(\mathbf{q}_n, \mathbf{K}, \mathbf{V}) = \frac{\sum_{m=1}^M g_m(\boldsymbol{\omega})}{\sum_{m=1}^M h_m(\boldsymbol{\omega})}, \tag{14}$$

$$\text{EVA}(\mathbf{q}_n, \mathbf{K}, \mathbf{V}) = \sum_{m \in E} \frac{\exp(\mathbf{q}_n^\top \mathbf{k}_m)}{Z} \frac{g_m(\boldsymbol{\omega})}{h_m(\boldsymbol{\omega})} + \sum_{c=1}^C \frac{\sum_{m \in \mathcal{P}_c} \exp(\mathbf{q}_n^\top \mathbf{k}_m)}{Z} \frac{\sum_{m \in \mathcal{P}_c} g_m(\boldsymbol{\omega})}{\sum_{m \in \mathcal{P}_c} h_m(\boldsymbol{\omega})}. \tag{15}$$

Intuitively, we can think of EVA as a calibrated version of RFA. Instead of directly computing and aggregating the random feature approximation for all tokens as in RFA (Equation 14), EVA (Equation 15) first constructs *local* estimation for either a single token ($m \in E$) or a subset (e.g., $\mathcal{P}_c$), and then *corrects* these approximations by their corresponding true attention scores (e.g., $\sum_{m \in \mathcal{P}_c} \exp(\mathbf{q}_n^\top \mathbf{k}_m)$ for $\mathcal{P}_c$). These adjusted local estimates are finally aggregated and globally normalized. Thanks to the decomposed representation of RFA, we can realize this divide-and-conquer strategy in a principled manner, which imposes finer-grained control on the whole estimation accuracy and enjoys increased approximation fidelity.

Table 1: Classification accuracy on `ImageNet1k` in comparison to different RF-based approximations. [†]vanilla PVT-v2-b3 (Wang et al., 2021b) uses a convolutional kernel to downsample key and value vectors, resulting in fewer FLOPs but with significant performance degradation.

| Model | DeiT-Tiny | | | DeiT-Small | | | PVT-v2-b3 | | |
|---|---|---|---|---|---|---|---|---|---|
| | # Param. | FLOPs | Top-1 Acc. | # Param. | FLOPs | Top-1 Acc. | # Param. | FLOPs | Top-1 Acc. |
| Local | 5.7M | 1.1G | 67.10 | 22.0M | 4.3G | 74.06 | 36.0M | 7.2G | 83.34 |
| Performer | 5.7M | 1.2G | 65.92 | 22.0M | 4.4G | 74.29 | 36.0M | 8.2G | 82.40 |
| LARA | 5.8M | 1.2G | 71.48 | 22.2M | 4.5G | 79.48 | 39.9M | 7.7G | 83.47 |
| EVA (Ours) | 5.8M | 1.2G | **73.00** | 22.2M | 4.4G | **80.65** | 36.1M | 7.4G | **83.71** |
| Softmax | 5.7M | 1.3G | **72.98** | 22.0M | 4.6G | 80.36 | 45.2M | 6.9G[†] | 83.14[†] |

## 4.2 PRACTICAL IMPLEMENTATION

According to the formulation (Equation 13) of EVA, the terms within $E$ could be computed efficiently due to its limited size; however, the partitioning requires computing $\sum_{m \in \mathcal{P}_c} \exp(\mathbf{q}_n^\top \mathbf{k}_m)$ explicitly for each subset, which again builds up to quadratic computational complexity. As discussed above, $\sum_{m \in \mathcal{P}_c} \exp(\mathbf{q}_n^\top \mathbf{k}_m)$ serves as a weight to correct the contribution from each subset $\mathcal{P}_c$. In this regard, we propose to approximate such control by $\sum_{m \in \mathcal{P}_c} \exp(\mathbf{q}_n^\top \mathbf{k}_m) \approx \exp(\mathbf{q}_n^\top \widetilde{\mathbf{k}}_c)$, where $\widetilde{\mathbf{k}}_c$ is an adaptive vector summarizing the information of all keys belonging to $\mathcal{P}_c$ (see Appendix C for more details). Such heuristic not only avoids computing the exponential dot product of each query-key pair explicitly, but also induces a fast approximation of the normalizing constant,

$$Z = \sum_{m \in E} \exp(\mathbf{q}_n^\top \mathbf{k}_m) + \sum_{c=1}^{C} \sum_{m \in \mathcal{P}_c} \exp(\mathbf{q}_n^\top \mathbf{k}_m) \approx \sum_{m \in E} \exp(\mathbf{q}_n^\top \mathbf{k}_m) + \sum_{c=1}^{C} \exp(\mathbf{q}_n^\top \widetilde{\mathbf{k}}_c).$$

Equipped with these results, our EVA estimator (Equation 13) can be reduced as follows,

$$\text{EVA}(\mathbf{q}_n, \mathbf{K}, \mathbf{V}) \approx \frac{\sum_{m \in E} \exp(\mathbf{q}_n^\top \mathbf{k}_m) \mathbf{v}_m + \sum_{c=1}^{C} \exp(\mathbf{q}_n^\top \widetilde{\mathbf{k}}_c) \widehat{\boldsymbol{\beta}}_c(\boldsymbol{\omega})}{\sum_{m \in E} \exp(\mathbf{q}_n^\top \mathbf{k}_m) + \sum_{c=1}^{C} \exp(\mathbf{q}_n^\top \widetilde{\mathbf{k}}_c)}. \quad (16)$$

**Parameterization Details.** We define $E$ in the same way as a simple block-wise local attention (Xiong et al., 2021a). The input sequence is first chunked into multiple blocks (or 2D windows for images), and each query $\mathbf{q}_n$ is associated with a specific $E^n$ that only contains tokens within the same block as the query. For the remaining indices $[M] \setminus E^n$, we evenly split it into $C$ contiguous chunks $\{\mathcal{P}_1^n, \dots, \mathcal{P}_C^n\}$. Note that we add the superscript $n$ here to denote the dependence on the query position; however, for notational brevity, we omit the notation when there is no ambiguity. The pseudo-code of EVA is provided in Algorithm 1 of Appendix. More implementation details, including the definition of $\widetilde{\mathbf{k}}_c$ and $\widehat{\boldsymbol{\beta}}_c(\boldsymbol{\omega})$ in Equation 16, are deferred to Appendix C.

**Extension to Autoregressive Modeling.** The decoder (or causal) self-attention, where each query can only attend to previous tokens, is the key ingredient in Transformer-based generative modeling (Vaswani et al., 2017; Brown et al., 2020). We demonstrate that it is straightforward to extend EVA to support such auto-regressive modeling with few modifications. Thanks to the decomposed formulation of EVA, we only need to incorporate two triangular mask matrices into the computation, which eliminate the information from future singletons $m \in E$ and entire future subsets $\mathcal{P}_c$ respectively. Unlike previous RFA methods, which are slow during training due to their recurrent computation (Choromanski et al., 2021; Peng et al., 2021b), the resulting causal variant remains highly efficient. More details can be found in Appendix D, including a pseudo-code Algorithm 2.

## 5 EXPERIMENTAL RESULTS

In this section, we evaluate our proposed method on various tasks, including image classification (§5.1), language tasks (§5.2), and Long Range Arena benchmark (Appendix F). Details of experimental protocols and baselines can be found in Appendix E.

Table 2: Image classification accuracy on `ImageNet1k` dataset with DeiT-Tiny-784.

| Model | # Param. | FLOPs | Top-1 Acc. |
|---|---|---|---|
| Performer (Choromanski et al., 2021) | 5.7M | 4.9G | 67.19 |
| Local attention (Child et al., 2019) | 5.7M | 4.4G | 70.62 |
| Scatterbrain (Chen et al., 2021a) | 5.7M | 5.2G | 73.50 |
| Nyströmformer (Xiong et al., 2021b) | 5.7M | 4.8G | 74.20 |
| LARA (Zheng et al., 2022b) | 5.8M | 4.6G | 75.02 |
| Combiner (Ren et al., 2021) | 5.7M | 4.7G | 75.56 |
| Long-Short (Zhu et al., 2021) | 6.1M | 5.0G | 76.41 |
| EVA (Ours) | 5.8M | 4.6G | 76.67 |
| Softmax | 5.7M | 7.0G | **77.16** |

Table 3: Masked Language Modeling Perplexity on the `Books3` validation dataset.

| Model | # Param. | FLOPs | Perplexity |
|---|---|---|---|
| Performer (Choromanski et al., 2021) | 126M | 213G | 8.61 |
| Linformer (Wang et al., 2020) | 129M | 193G | 5.16 |
| LARA (Zheng et al., 2022b) | 126M | 194G | 4.39 |
| Reformer (Kitaev et al., 2020) | 126M | 205G | 4.28 |
| Local attention (Child et al., 2019) | 136M | 183G | 4.27 |
| Combiner (Ren et al., 2021) | 136M | 187G | 4.12 |
| Long-Short (Zhu et al., 2021) | 142M | 218G | 4.01 |
| EVA (Ours) | 136M | 184G | 3.94 |
| EVA-4096 (Ours) | 136M | 387G | **3.73** |
| Softmax | 126M | 252G | **3.74** |

Table 4: BLEU scores on the test set of `WMT14 En-De`. [†] numbers are taken from Zheng et al. (2022b).

| Model | # Param. | BLEU |
|---|---|---|
| Performer-128[†] | 60.92M | 23.5 |
| LARA-16[†] | 60.96M | 26.4 |
| LARA-32[†] | 60.96M | 26.8 |
| LARA-64[†] | 60.96M | 27.0 |
| EVA-16 | 60.96M | 27.2 |
| EVA-32 | 60.96M | 27.3 |
| EVA-64 | 60.96M | **27.5** |
| Softmax | 60.92M | **27.5** |

Table 5: Validation (Val.) and Test perplexity (PPL) on `Wikitext-103`. 256/480 indicate evaluation context window sizes. [†] numbers are due to Kasai et al. (2021).

| Model | # Params. | 256 | | 480 | |
|---|---|---|---|---|---|
| | | Val. | Test | Val. | Test |
| Softmax[†] | 449M | **17.9** | **18.5** | – | – |
| ELU[†] | 449M | 22.0 | 22.8 | – | – |
| RFA[†] | 449M | 20.4 | 21.3 | – | – |
| T2R[†] | 450M | 20.1 | 20.8 | – | – |
| EVA (Ours) | 450M | **17.9** | 18.6 | **17.7** | **18.3** |
| Softmax | 247M | 18.8 | 19.5 | 18.4 | 19.1 |
| EVA (Ours) | 247M | 18.8 | 19.4 | 18.5 | 19.1 |

## 5.1 IMAGE CLASSIFICATION

We explore the ability to learn visual representations for different attention mechanisms in vision transformers (ViTs; Dosovitskiy et al., 2021). In particular, we replace softmax attention used in ViTs with its efficient variants and evaluate their performance on the `ImageNet1k` dataset (Deng et al., 2009), which contains over 1,280K and 50K images of 1,000 classes for training and validation splits, respectively. For the transformer model, we consider both a plain ViT (DeiT; Dosovitskiy et al., 2020; Touvron et al., 2021) and a pyramidal ViT (PVT; Wang et al., 2021b) to test the performance. The former maintains the same sequence length (which is set to 196 by default) across all transformer layers, while the latter processes much longer sequences (up to 3136 tokens) at early layers and progressively reduces the sequence length to form a hierarchical structure. Detailed experimental settings could be found in Appendix E.2.

**Results.** We first compare the performance of EVA against our main baselines on the standard ViT architectures. As shown in Table 1, EVA significantly improves the performance of previous RFA approaches (including Performer (Choromanski et al., 2021) and LARA (Zheng et al., 2022b)) and local attention by a large margin, and even outperforms the conventional softmax attention. We then consider a more challenging setting, where the plain architecture DeiT-Tiny is used but the sequence length is scaled up to 784 (denoted as DeiT-Tiny-784). We compare EVA against other attention variants in this setting and report the classification results in Table 2. EVA outperforms most previous baselines and remains highly competitive with softmax attention, illustrating its effectiveness.

## 5.2 MACHINE TRANSLATION AND LANGUAGE MODELING

We further evaluate EVA on the natural language domain. Specifically, we consider three tasks:

- **Masked language modeling (MLM)** on a pretraining-scale book corpus `Books3` in the Pile dataset suite (Presser, 2020; Gao et al., 2020), consisting of over 196,640 published books.
- **Machine translation (MT)** on `WMT14 En-De` benchmark (Bojar et al., 2014).
- **Autoregressive language modeling (Autoregressive LM)** on a large-scale token-level LM benchmark `Wikitext-103` (Merity et al., 2016).

**Results.** We report **MLM** validation perplexity in Table 3, where the sequence length is 2048 by default. EVA substantially improves previous methods based on random features (including Performer and LARA) and outperforms the other efficient attention mechanisms. Thanks to the linear

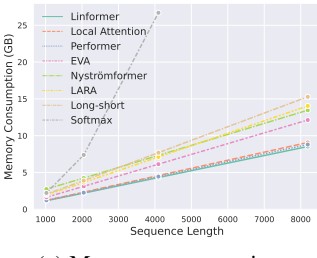

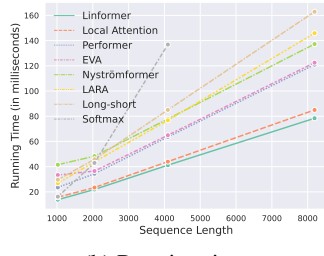

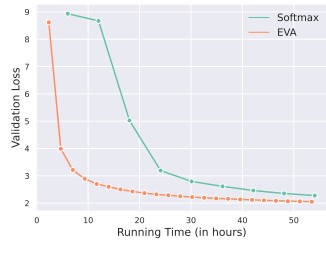

(a) Memory consumption.  (b) Running time.  (c) Training speed-up.

Figure 1: **Left** and **middle**: empirical memory consumption and running time comparison respectively of different attention mechanisms under various sequence lengths. **Right**: a snapshot of MLM validation loss curve versus actual elapsed time during training.

complexity of EVA, it can be scaled further to much longer sequences. With input sequences of length increased to 4096, EVA (denoted as "EVA-4096") attains lower validation perplexity than exact softmax attention, which demonstrates its capability of scaling to much longer sequences. Besides, **machine translation** results are compared in Table 4, where in this task $C = 8$ by default and EVA-$m$ denotes EVA with $|E| = m$. EVA outperforms previous random feature methods by a large margin and achieves translation quality on par with full softmax attention even under the setting of small $|E|$ and $C$. For **Autoregressive LM** (Table 5), EVA achieves the same perplexity as softmax attention with much lower computational complexity. Comparing against various random feature methods reported by previous work Kasai et al. (2021), we observe a significant performance gain brought from EVA even under a Transformer with half parameters. When further increasing the transformer model size as the setting in Kasai et al. (2021), EVA still scales as effectively as softmax attention with a comparable perplexity while outperforming previous random feature methods by a larger margin. These results indicate the substantially enlarged capacity of EVA to approximate softmax attention.

### 5.3 ANALYSIS

**Running Time & Memory Comparison.** We conduct a simulation experiment to evaluate the empirical efficiency of various attention methods, which is measured by the running time per iteration and memory footprint under different sequence lengths. The setup can be found in Appendix E.4. As illustrated in Figures 1a and 1b, EVA only incurs a little computational overhead compared to Performer and local attention and achieves much better running time speed-up than Long-Short (Zhu et al., 2021), a strong baseline across various tasks albeit with much longer running time and larger memory consumption. In Figure 1c, we further visualize the speed-up of EVA relative to conventional softmax attention by plotting the validation loss curve versus actual elapsed time during training transformers (equivalent to 32 GPU days). It can be seen that EVA can achieve a much lower loss after running for the same elapsed time; in contrast, conventional softmax attention needs to run almost $3\times$ longer to match the loss quantity. Overall, our method attains a good trade-off between quality and empirical efficiency.

Table 6: Classification accuracy on `ImageNet1k` dataset.

| Model | Mem.(G) | Time(ms/iter) | $|E|$ | $C$ | Top-1 Acc. |
|---|---|---|---|---|---|
| Performer | 8.1 | 87 | 0 | 1 | 67.19 |
| Local | 7.8 | 65 | 49 | 0 | 70.62 |
| | 8.4 | 77 | 0 | 49 | 74.33 |
| | 9.1 | 87 | 49 | 1 | 74.10 |
| EVA | 9.4 | 89 | 49 | 16 | 75.83 |
| | 9.9 | 94 | 49 | 49 | 76.67 |
| | 12.5 | 119 | 49 | 196 | 77.10 |
| | 11.9 | 108 | 196 | 49 | 77.36 |
| Softmax | 17.7 | 99 | n.a. | n.a. | 77.16 |

Table 7: MLM validation perplexity on `Books3`. "–" indicates fail to converge.

| Model | Mem.(G) | Time(ms/iter) | $|E|$ | $C$ | Perplexity |
|---|---|---|---|---|---|
| Performer-4096 | 4.8 | 39 | 0 | 1 | – |
| Local-4096 | 4.4 | 29 | 256 | 0 | 4.34 |
| | 5.8 | 40 | 256 | 128 | 3.82 |
| EVA-4096 | 6.4 | 41 | 256 | 256 | 3.73 |
| | 6.9 | 47 | 512 | 128 | 3.71 |
| Softmax-4096 | 21.2 | 102 | n.a. | n.a. | 3.65 |

**Ablation Study.** In this section, we conduct an ablation study on image classification and MLM tasks to investigate the effects of main hyper-parameters in EVA (see Table 8 for more comprehensive analysis). In particular, we vary $|E|$ and the partition size $C$ and evaluate their performance on both image classification and masked language modeling. As presented in Table 6 and Table 7, increasing $|E|$ amounts to obtaining exact estimates for more key-value pairs, which greatly improves empirical performance; besides, increasing $C$ would process control variates at a finer scale, also translating into better modeling quality, consistent with our theoretical analysis (§4.1).

## 6 RELATED WORK

**Control Variates.** Control variates are a widely used variance reduction technique in reinforcement learning (Greensmith et al., 2004; Grathwohl et al., 2018; Vlassis et al., 2021), stochastic optimization (Wang et al., 2013), variational inference (Paisley et al., 2012; Ranganath et al., 2014; Geffner & Domke, 2018; Tucker et al., 2017; Grathwohl et al., 2018), Markov chain Monte Carlo (Baker et al., 2019) and many other topics. Our construction with control variates provides a new perspective on designing faster yet more accurate attention approximations.

**Efficient Attention Mechanisms.** A lot of research work has put the focus on reducing the quadratic complexity of conventional softmax attention. A widely used approach is to define a sparse attention pattern so that each query is limited to only attending to a subset of tokens. The sparse pattern could be either learnable (Kitaev et al., 2020; Vyas et al., 2020; Tay et al., 2020; Roy et al., 2021; Madaan et al., 2022) or simply fixed (Liu et al., 2018; Parmar et al., 2018; Child et al., 2019; Beltagy et al., 2020; Ainslie et al., 2020; Zaheer et al., 2020; Liu et al., 2021; Xiong et al., 2021a; Wang et al., 2022; Chen et al., 2022; Hutchins et al., 2022). Another paradigm is to adopt low-rank approximations, including via the Nyström method (Xiong et al., 2021b), down-sampling with learnable projections (Wang et al., 2020; Peng et al., 2021a), or explicitly compressing sequences (Rae et al., 2020; Dai et al., 2020; Ma et al., 2021; Jaegle et al., 2021). There are also studies improving both sparse and low-rank methods for better attention matrix approximation (Nguyen et al., 2021; Zhu et al., 2021; Chen et al., 2021a; Ren et al., 2021; Zhu & Soricut, 2021; Hua et al., 2022; Zeng et al., 2022). Instead of adopting approximate methods, a recent line of work (Rabe & Staats, 2021; Dao et al., 2022) proposes to compute the exact softmax attention in an online manner (Milakov & Gimelshein, 2018) without materializing the full attention matrix. In this way, softmax attention can be computed in linear memory complexity, and the runtime can also be greatly improved by further minimizing memory accesses (Dao et al., 2022).

**Random-Feature-based Attention.** Random-feature-based methods are a popular alternative that uses random features (Rahimi & Recht, 2008) to linearize exponential kernels in softmax attention (Katharopoulos et al., 2020; Choromanski et al., 2021; Peng et al., 2021b). Recent work attempts to improve RFA approximation from several aspects, such as designing more accurate random feature maps (Choromanski et al., 2022; Likhosherstov et al., 2022; Chowdhury et al., 2022), incorporating relative positional or other task-specific biases (Liutkus et al., 2021; Luo et al., 2021; Chen, 2021; Zheng et al., 2022a; Qin et al., 2022b; Wu et al., 2022; Qin et al., 2022a), or leveraging connections to fast weight programmers (Peng et al., 2021b; Schlag et al., 2021; Irie et al., 2021). Prior work closely related to ours includes Zheng et al. (2022b), which reinterprets RFA using self-normalized importance sampling (Hesterberg, 1995) and theoretically extends the random feature approximation from individual exponential kernels to the whole softmax attention. Our work further generalizes this result via control variates and characterizes the approximation gap caused by RFA. Scatterbrain (Chen et al., 2021a) is also similar to our work in that it also refines RF approximation on critical local regions. However, it is developed based on a different motivation that attempts to approximate the attention matrix with a combination of sparse and low-rank matrices. Interestingly, we find that Scatterbrain can be cast as a special case under our framework; see Appendix G for a detailed discussion about connections between EVA and previous attention mechanisms.

## 7 CONCLUSION AND LIMITATIONS

In this work, we develop an efficient attention mechanism EVA via control variates. Our framework reveals a localized perspective of RFA approximation, which not only bridges the gap between RFA and exact softmax attention but also attains a good trade-off between modeling quality and efficiency. We evaluate our method on both vision and language tasks and demonstrate substantial improvements over previous baselines. There are some limitations of our framework. For instance, the approximation in computing control variate estimation for each partitioned subset is crude and might limit the potential modeling capacity; in addition, we only explore the most straightforward partitioning strategy that evenly splits the sequence into multiple contiguous chunks; while in general, the partition could contain arbitrary subsequences or be adaptive to inputs via clustering methods, which can be guided by task-specific inductive biases. It is interesting to investigate these limitations to unleash the expressiveness of EVA further, which we leave for future work.

ACKNOWLEDGMENTS

We would like to thank the HKU NLP group, the Shark-NLP group, and the anonymous reviewers for their valuable suggestions that greatly helped improve this work. This work is partially supported by the joint research scheme of the National Natural Science Foundation of China (NSFC) and the Research Grants Council (RGC) under grant number N_HKU714/21.

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

# Appendix

# A  A BRIEF REVIEW OF VANILLA RANDOM FEATURE ATTENTION

Vanilla random feature attention methods, such as Performer (Choromanski et al., 2021; Peng et al., 2021b), seek to approximate the softmax attention mechanism through *random features* (Rahimi & Recht, 2008) $\phi(\mathbf{x}, \boldsymbol{\omega}) := 1/\sqrt{S}[\xi(\mathbf{x}, \omega_1), \ldots, \xi(\mathbf{x}, \omega_S)]^\top$. Here, $\omega_1, \ldots, \omega_S \sim \mathcal{N}(0, \mathbf{I})$, and $\xi(\mathbf{x}, \omega)$ is the *randomized mapping* such that

$$\exp\left(\mathbf{q}_n^\top \mathbf{k}_m\right) = \mathbb{E}_{\omega_s \sim \mathcal{N}(0, \mathbf{I})}\left[\xi(\mathbf{q}_n, \omega_s)^\top \xi(\mathbf{k}_m, \omega_s)\right]. \tag{17}$$

Therefore, we can draw multiple Monte Carlo samples to estimate the exponential kernel,

$$\exp\left(\mathbf{q}_n^\top \mathbf{k}_m\right) \approx \frac{1}{S}\sum_{s=1}^{S} \xi(\mathbf{q}_n, \omega_s)^\top \xi(\mathbf{k}_m, \omega_s) := \phi(\mathbf{q}_n, \boldsymbol{\omega})^\top \phi(\mathbf{k}_m, \boldsymbol{\omega}),$$

and then approximate the attention mechanism as

$$\sum_{m=1}^{M} \frac{\exp\left(\mathbf{q}_n^\top \mathbf{k}_m\right)}{\sum_{m'=1}^{M} \exp\left(\mathbf{q}_n^\top \mathbf{k}_{m'}\right)}\mathbf{v}_m \approx \frac{\sum_{m=1}^{M} \phi(\mathbf{q}_n, \boldsymbol{\omega})^\top \phi(\mathbf{k}_m, \boldsymbol{\omega})\mathbf{v}_m}{\sum_{m'=1}^{M} \phi(\mathbf{q}_n, \boldsymbol{\omega})^\top \phi(\mathbf{k}_{m'}, \boldsymbol{\omega})}. \tag{18}$$

It is recently generalized as a self-normalized importance sampling estimator to approximate softmax attention (Zheng et al., 2022b), as described in §2.2. We refer the generalized random feature based approximations as RFA.

# B  PROOFS & DERIVATIONS

## B.1  AN EXTENDED REVIEW OF CONTROL VARIATES

The control variate method takes the following form,

$$\widetilde{g}(\boldsymbol{\omega}) = g(\boldsymbol{\omega}) - \boldsymbol{\beta}h(\boldsymbol{\omega}) + \boldsymbol{\beta}\mathbb{E}\left[h(\boldsymbol{\omega})\right], \tag{19}$$

Given the particular forms of $g(\cdot)$ and $h(\cdot)$, $\boldsymbol{\beta}$ can be optimized to minimize the estimation variance. For notational convenience, we denote the covariance between a scalar and a random vector as $\mathrm{Cov}\left[h(\boldsymbol{\omega}), g(\boldsymbol{\omega})\right] := \mathbb{E}\left[(h(\boldsymbol{\omega}) - \mathbb{E}\left[h(\boldsymbol{\omega})\right])(g(\boldsymbol{\omega}) - \mathbb{E}\left[g(\boldsymbol{\omega})\right])\right]$, and the variance of a random vector as $\mathrm{Var}\left[g(\boldsymbol{\omega})\right] := \mathrm{Cov}\left[g(\boldsymbol{\omega}), g(\boldsymbol{\omega})\right]$. In particular, we have

$$\begin{aligned}
\mathrm{Var}\left[\widetilde{g}(\boldsymbol{\omega})\right] &= \mathrm{Var}\left[g(\boldsymbol{\omega}) - \boldsymbol{\beta}h(\boldsymbol{\omega})\right] \\
&= \mathrm{Var}\left[g(\boldsymbol{\omega})\right] - 2\,\mathrm{Cov}\left[\boldsymbol{\beta}h(\boldsymbol{\omega}), g(\boldsymbol{\omega})\right] + \mathrm{Var}\left[\boldsymbol{\beta}h(\boldsymbol{\omega})\right] \\
&= \mathrm{Var}\left[g(\boldsymbol{\omega})\right] - 2\,\mathrm{Cov}\left[h(\boldsymbol{\omega}), g(\boldsymbol{\omega})\right]\boldsymbol{\beta}^\top + \mathrm{Var}\left[h(\boldsymbol{\omega})\right]\boldsymbol{\beta}\boldsymbol{\beta}^\top.
\end{aligned}$$

We hope an optimal $\boldsymbol{\beta}$ would minimize $\mathrm{Tr}\left(\mathrm{Var}\left[\widetilde{g}(\boldsymbol{\omega})\right]\right)$, that is, the sum of estimating variance for each dimension. By differentiating, we obtain

$$\boldsymbol{\beta}^* = \arg\min_{\boldsymbol{\beta}} \mathrm{Tr}\left(\mathrm{Var}\left[\widetilde{g}(\boldsymbol{\omega})\right]\right) = \frac{\mathrm{Cov}\left[h(\boldsymbol{\omega}), g(\boldsymbol{\omega})\right]}{\mathrm{Var}\left[h(\boldsymbol{\omega})\right]}. \tag{20}$$

Since both the covariance and the variance may be intractable to compute, the optimal $\boldsymbol{\beta}^*$ is generally not available in closed form. Nevertheless, with the optimal coefficient, the variance of such control variate estimate would never be larger than the plain estimator $g(\cdot)$.

## B.2  DERIVATION OF SNIS AS CONTROL VARIATE ESTIMATION

For notational convenience, we denote the importance weight as $W(\omega_s) := p_n(\omega_s)/q(\omega_s)$. Then we have

$$
\begin{aligned}
\widetilde{g}(\boldsymbol{\omega}) &= \frac{\sum_{s=1}^{S} \frac{p_n(\omega_s)}{q(\omega_s)} f(\omega_s)}{\sum_{s=1}^{S} \frac{p_n(\omega_s)}{q(\omega_s)}} = \frac{\sum_{s=1}^{S} W(\omega_s) f(\omega_s)}{\sum_{s=1}^{S} W(\omega_s)} \\
&= \frac{\sum_{s=1}^{S} W(\omega_s) f(\omega_s)}{\sum_{s=1}^{S} W(\omega_s)} - \frac{1}{S} \sum_{s=1}^{S} W(\omega_s) f(\omega_s) + \frac{1}{S} \sum_{s=1}^{S} W(\omega_s) f(\omega_s) \\
&= \frac{\sum_{s=1}^{S} W(\omega_s) f(\omega_s)}{\sum_{s=1}^{S} W(\omega_s)} - \frac{\sum_{s=1}^{S} W(\omega_s)}{\sum_{s=1}^{S} W(\omega_s)} \frac{1}{S} \sum_{s=1}^{S} W(\omega_s) f(\omega_s) + \frac{1}{S} \sum_{s=1}^{S} W(\omega_s) f(\omega_s) \\
&= \frac{1 - \frac{1}{S} \sum_{s=1}^{S} W(\omega_s)}{\sum_{s=1}^{S} W(\omega_s)} \sum_{s=1}^{S} W(\omega_s) f(\omega_s) + \frac{1}{S} \sum_{s=1}^{S} W(\omega_s) f(\omega_s) \\
&= \frac{\sum_{s=1}^{S} W(\omega_s) f(\omega_s)}{\sum_{s=1}^{S} W(\omega_s)} \left( 1 - \frac{1}{S} \sum_{s=1}^{S} W(\omega_s) \right) + \frac{1}{S} \sum_{s=1}^{S} W(\omega_s) f(\omega_s) \\
&= \frac{1}{S} \sum_{s=1}^{S} W(\omega_s) f(\omega_s) - \frac{\sum_{s=1}^{S} W(\omega_s) f(\omega_s)}{\sum_{s=1}^{S} W(\omega_s)} \left( \frac{1}{S} \sum_{s=1}^{S} W(\omega_s) - 1 \right) \\
&= g(\boldsymbol{\omega}) - \widehat{\boldsymbol{\beta}}(\boldsymbol{\omega}) \left( h(\boldsymbol{\omega}) - \mathbb{E}\left[ h(\boldsymbol{\omega}) \right] \right),
\end{aligned}
$$

Note that the expectation of importance weights equals 1, that is,

$$
\mathbb{E}\left[ h(\boldsymbol{\omega}) \right] = \mathbb{E}\left[ \frac{1}{S} \sum_{s=1}^{S} W(\omega_s) \right] = \mathbb{E}_{\omega_1,\ldots,\omega_S \sim q(\omega)} \left[ \sum_{s=1}^{S} \frac{1}{S} \frac{p(\omega_s)}{q(\omega_s)} \right] = \frac{1}{S} \sum_{s=1}^{S} \mathbb{E}_{\omega_s \sim q(\omega)} \left[ \frac{p(\omega_s)}{q(\omega_s)} \right] = 1.
$$

Same as SNIS, this estimator is still biased due to the dependence of $\widehat{\boldsymbol{\beta}}(\boldsymbol{\omega})$ on $\boldsymbol{\omega}$. However, it would asymptotically become unbiased since $\widehat{\boldsymbol{\beta}}(\boldsymbol{\omega})$ is consistent and converges to a constant $\boldsymbol{\beta}$ w.r.t. $\omega$ given a large number of samples,

$$
\widehat{\boldsymbol{\beta}}(\boldsymbol{\omega}) = \frac{g(\boldsymbol{\omega})}{h(\boldsymbol{\omega})} \xrightarrow{p} \frac{\mathbb{E}\left[ g(\boldsymbol{\omega}) \right]}{\mathbb{E}\left[ h(\boldsymbol{\omega}) \right]} = \underbrace{\mathbb{E}_{p_n(\omega)}\left[ f(\omega) \right]}_{\text{constant}} := \boldsymbol{\beta}. \tag{21}
$$

## B.3  DERIVATION OF THE EXPECTATION OF PER-TERM CONTROL VARIATES

According to the definition of randomized mappings, we have

$$
\begin{aligned}
\mathbb{E}\left[ h_m(\boldsymbol{\omega}) \right] &= \mathbb{E}_{\omega_1,\ldots,\omega_S \sim q(\omega)} \left[ \frac{1}{S} \sum_{s=1}^{S} \frac{\mathcal{N}(\omega_s; 0, \mathbf{I})}{Z q(\omega_s)} \xi(\mathbf{q}_n, \omega_s) \xi(\mathbf{k}_m, \omega_s) \right] \\
&= \frac{1}{S} \sum_{s=1}^{S} \frac{1}{Z} \int \xi(\mathbf{q}_n, \omega_s) \xi(\mathbf{k}_m, \omega_s) \mathcal{N}(\omega_s; 0, \mathbf{I}) d\omega_s \\
&= \frac{\exp(\mathbf{q}_n^\top \mathbf{k}_m)}{Z}. \tag{22}
\end{aligned}
$$

## B.4  PROOF OF PROPOSITION 1

*Proof.* We start with the formulation of $g(\cdot)$ and $h(\cdot)$,

$$
\frac{g_m(\boldsymbol{\omega})}{h_m(\boldsymbol{\omega})} = \frac{\sum_{s=1}^{S} \frac{\mathcal{N}(\omega_s; 0, \mathbf{I})}{Z q(\omega_s)} \xi(\mathbf{q}_n, \omega_s) \xi(\mathbf{k}_m, \omega_s) \mathbf{v}_m}{\sum_{s=1}^{S} \frac{\mathcal{N}(\omega_s; 0, \mathbf{I})}{Z q(\omega_s)} \xi(\mathbf{q}_n, \omega_s) \xi(\mathbf{k}_m, \omega_s)} = \mathbf{v}_m.
$$

As a result, we have $g_m(\boldsymbol{\omega}) = h_m(\boldsymbol{\omega})\mathbf{v}_m$ and $\mathbb{E}[g_m(\boldsymbol{\omega})] = \mathbb{E}[h_m(\boldsymbol{\omega})]\mathbf{v}_m$. We now investigate the optimal $\boldsymbol{\beta}_m$ according to Equation 20,

$$
\begin{aligned}
\boldsymbol{\beta}_m^* &= \arg\min_{\boldsymbol{\beta}} \mathrm{Tr}\left(\mathrm{Var}\left[\widetilde{g}_m(\boldsymbol{\omega})\right]\right) \\
&= \frac{\mathrm{Cov}\left[h_m(\boldsymbol{\omega}), g_m(\boldsymbol{\omega})\right]}{\mathrm{Var}\left[h_m(\boldsymbol{\omega})\right]} \\
&= \frac{\mathbb{E}\left[(h(\boldsymbol{\omega}) - \mathbb{E}[h(\boldsymbol{\omega})])(h(\boldsymbol{\omega}) - \mathbb{E}[h(\boldsymbol{\omega})])\right]\mathbf{v}_m}{\mathbb{E}\left[(h(\boldsymbol{\omega}) - \mathbb{E}[h(\boldsymbol{\omega})])(h(\boldsymbol{\omega}) - \mathbb{E}[h(\boldsymbol{\omega})])\right]} \\
&= \mathbf{v}_m = \frac{g_m(\boldsymbol{\omega})}{h_m(\boldsymbol{\omega})}.
\end{aligned}
$$

In terms of the variance, we again use $g_m(\boldsymbol{\omega}) = h_m(\boldsymbol{\omega})\mathbf{v}_m$ to obtain

$$
\begin{aligned}
\widetilde{g}_m(\boldsymbol{\omega}) &= g_m(\boldsymbol{\omega}) - \widehat{\boldsymbol{\beta}}_m\left(h_m(\boldsymbol{\omega}) - \mathbb{E}[h_m(\boldsymbol{\omega})]\right) \\
&= g_m(\boldsymbol{\omega}) - \mathbf{v}_m h_m(\boldsymbol{\omega}) + \mathbf{v}_m\mathbb{E}[h_m(\boldsymbol{\omega})] \\
&= \mathbf{v}_m\mathbb{E}[h_m(\boldsymbol{\omega})] \\
&= \frac{\exp(\mathbf{q}_n^\top\mathbf{k}_m)}{Z}\mathbf{v}_m.
\end{aligned}
\tag{23}
$$

Since this holds true for every term $m = 1, \ldots, M$, our estimate becomes exactly softmax attention,

$$
\widetilde{g}(\boldsymbol{\omega}) = \sum_{m=1}^{M}\widetilde{g}_m(\boldsymbol{\omega}) = \sum_{m=1}^{M}\frac{\exp(\mathbf{q}_n^\top\mathbf{k}_m)}{Z}\mathbf{v}_m = \sum_{m=1}^{M}\frac{\exp(\mathbf{q}_n^\top\mathbf{k}_m)}{\sum_{m'=1}^{M}\exp(\mathbf{q}_n^\top\mathbf{k}_m)}\mathbf{v}_m.
$$

Since all randomness is eliminated, the estimate is exact with zero bias and variance. That is, $\mathrm{RFA}(\mathbf{q}_n, \mathbf{K}, \mathbf{V}) = \widetilde{g}(\boldsymbol{\omega}) = \mathsf{SoftmaxAttn}(\mathbf{q}_n, \mathbf{K}, \mathbf{V})$. □

## B.5 A Formal Analysis of the Advantage of Partitioning

In this section, we demonstrate the advantage of partitioning by showing that there always exists some set $\{\boldsymbol{\beta}_c\}_{c=1}^C$ that achieves lower weighted MSE than any globally shared coefficient, as discussed in §4.1.

**Lemma 3.** *Suppose $\boldsymbol{\beta}_m^*$ is the optimal coefficient for each $m \in [M]$ as defined in Proposition 1, and $\mathcal{P}_1, \mathcal{P}_2, \ldots, \mathcal{P}_C$ are an arbitrary partition of $[M]$, where each subset $\mathcal{P}_c$ is associated with a distinct $\boldsymbol{\beta}_c$. We consider the following weighted mean squared error,*

$$
J(\boldsymbol{\beta}_1, \ldots, \boldsymbol{\beta}_C) := \sum_{c=1}^{C}\sum_{m\in\mathcal{P}_c}\alpha_m\|\boldsymbol{\beta}_c - \boldsymbol{\beta}_m^*\|^2,
\tag{24}
$$

*where $\alpha_m > 0$ for each $m \in [M]$ and $\sum_{c=1}^{C}\sum_{m\in\mathcal{P}_c}\alpha_m = 1$. Then for any choice of $\{\alpha_m\}_{m=1}^M$ and any globally shared coefficient $\boldsymbol{\beta}$, there exists some $\{\boldsymbol{\beta}_c^*\}_{c=1}^C$ so that*

$$
J(\boldsymbol{\beta}_1 = \boldsymbol{\beta}, \ldots, \boldsymbol{\beta}_C = \boldsymbol{\beta}) \geq J(\boldsymbol{\beta}_1 = \boldsymbol{\beta}_1^*, \ldots, \boldsymbol{\beta}_C = \boldsymbol{\beta}_C^*).
$$

*Proof.* Let $\boldsymbol{\beta}_c^* = \frac{\sum_{m\in\mathcal{P}_c}\alpha_m\boldsymbol{\beta}_m^*}{\sum_{m\in\mathcal{P}_c}\alpha_m}$ for each $c = 1, \ldots, C$. Then we have

$$
\begin{aligned}
\sum_{m\in\mathcal{P}_c}\alpha_m(\boldsymbol{\beta}_c^* - \boldsymbol{\beta}_m^*) &= \boldsymbol{\beta}_c^*\left(\sum_{m\in\mathcal{P}_c}\alpha_m\right) - \sum_{m\in\mathcal{P}_c}\alpha_m\boldsymbol{\beta}_m^* \\
&= \frac{\sum_{m\in\mathcal{P}_c}\alpha_m\boldsymbol{\beta}_m^*}{\sum_{m\in\mathcal{P}_c}\alpha_m}\left(\sum_{m\in\mathcal{P}_c}\alpha_m\right) - \sum_{m\in\mathcal{P}_c}\alpha_m\boldsymbol{\beta}_m^* \\
&= \sum_{m\in\mathcal{P}_c}\alpha_m\boldsymbol{\beta}_m^* - \sum_{m\in\mathcal{P}_c}\alpha_m\boldsymbol{\beta}_m^* = 0.
\end{aligned}
\tag{25}
$$

According to Equations 25 and 24, for any $\boldsymbol{\beta}$ we have the following inequality,

$$
\begin{aligned}
& J(\boldsymbol{\beta}_1 = \boldsymbol{\beta}, \ldots, \boldsymbol{\beta}_C = \boldsymbol{\beta}) \\
& = \sum_{c=1}^{C} \sum_{m \in \mathcal{P}_c} \alpha_m \left\| \boldsymbol{\beta} - \boldsymbol{\beta}_m^* \right\|^2 \\
& = \sum_{c=1}^{C} \sum_{m \in \mathcal{P}_c} \alpha_m \left\| \boldsymbol{\beta} - \boldsymbol{\beta}_c^* + \boldsymbol{\beta}_c^* - \boldsymbol{\beta}_m^* \right\|^2 \\
& = \sum_{c=1}^{C} \sum_{m \in \mathcal{P}_c} \alpha_m \left( \left\| \boldsymbol{\beta} - \boldsymbol{\beta}_c^* \right\|^2 + 2 \left( \boldsymbol{\beta} - \boldsymbol{\beta}_c^* \right)^\top \left( \boldsymbol{\beta}_c^* - \boldsymbol{\beta}_m^* \right) + \left\| \boldsymbol{\beta}_c^* - \boldsymbol{\beta}_m^* \right\|^2 \right) \\
& = \sum_{c=1}^{C} \sum_{m \in \mathcal{P}_c} \alpha_m \left\| \boldsymbol{\beta} - \boldsymbol{\beta}_c^* \right\|^2 + 2 \underbrace{\sum_{c=1}^{C} \sum_{m \in \mathcal{P}_c} \alpha_m \left( \boldsymbol{\beta} - \boldsymbol{\beta}_c^* \right)^\top \left( \boldsymbol{\beta}_c^* - \boldsymbol{\beta}_m^* \right)}_{=0} + \sum_{c=1}^{C} \sum_{m \in \mathcal{P}_c} \alpha_m \left\| \boldsymbol{\beta}_c^* - \boldsymbol{\beta}_m^* \right\|^2 \\
& = \sum_{c=1}^{C} \sum_{m \in \mathcal{P}_c} \alpha_m \left\| \boldsymbol{\beta} - \boldsymbol{\beta}_c^* \right\|^2 + \sum_{c=1}^{C} \sum_{m \in \mathcal{P}_c} \alpha_m \left\| \boldsymbol{\beta}_c^* - \boldsymbol{\beta}_m^* \right\|^2 \\
& \geq \sum_{c=1}^{C} \sum_{m \in \mathcal{P}_c} \alpha_m \left\| \boldsymbol{\beta}_c^* - \boldsymbol{\beta}_m^* \right\|^2 \\
& = J(\boldsymbol{\beta}_1 = \boldsymbol{\beta}_1^*, \ldots, \boldsymbol{\beta}_C = \boldsymbol{\beta}_C^*).
\end{aligned}
$$

As a result, for any choice of $\{\alpha_m\}_{m=1}^{M}$ and any globally shared coefficient $\boldsymbol{\beta}$, there always exists some $\{\boldsymbol{\beta}_c\}_{c=1}^{C}$ that achieves lower (or equal) weighted MSE, and a solution can be simply $\boldsymbol{\beta}_c = \frac{\sum_{m \in \mathcal{P}_c} \alpha_m \boldsymbol{\beta}_m^*}{\sum_{m \in \mathcal{P}_c} \alpha_m}$. $\qquad \square$

## B.6 Proof of Proposition 2

*Proof.* We first consider the case of partitioned indices, where each subset $\mathcal{P}_c$ is associated with some specific $\boldsymbol{\beta}_c$. To see the global minimum of $J$, we differentiate on both sides and obtain

$$
\begin{aligned}
\frac{\partial J(\boldsymbol{\beta}_1, \ldots, \boldsymbol{\beta}_C)}{\partial \boldsymbol{\beta}_c} & = \frac{\partial}{\partial \boldsymbol{\beta}_c} \sum_{c=1}^{C} \sum_{m \in \mathcal{P}_c} \frac{\exp \left( \mathbf{q}_n^\top \mathbf{k}_m \right)}{\sum_{m' \in U} \exp \left( \mathbf{q}_n^\top \mathbf{k}_{m'} \right)} \left\| \boldsymbol{\beta}_c - \boldsymbol{\beta}_m^* \right\|^2 \\
& = \sum_{m \in \mathcal{P}_c} \frac{\exp \left( \mathbf{q}_n^\top \mathbf{k}_m \right)}{\sum_{m' \in U} \exp \left( \mathbf{q}_n^\top \mathbf{k}_{m'} \right)} 2 \left( \boldsymbol{\beta}_c - \boldsymbol{\beta}_m^* \right).
\end{aligned}
$$

By setting the partial derivative to zero, we obtain

$$
\begin{aligned}
\boldsymbol{\beta}_c^* & = \frac{\sum_{m \in \mathcal{P}_c} \exp \left( \mathbf{q}_n^\top \mathbf{k}_m \right) \boldsymbol{\beta}_m^*}{\sum_{m \in \mathcal{P}_c} \exp \left( \mathbf{q}_n^\top \mathbf{k}_m \right)} = \frac{\sum_{m \in \mathcal{P}_c} \exp \left( \mathbf{q}_n^\top \mathbf{k}_m \right) \mathbf{v}_m}{\sum_{m \in \mathcal{P}_c} \exp \left( \mathbf{q}_n^\top \mathbf{k}_m \right)} \\
& = \frac{\sum_{m \in \mathcal{P}_c} \mathbb{E} \left[ g_m(\boldsymbol{\omega}) \right]}{\sum_{m \in \mathcal{P}_c} \mathbb{E} \left[ h_m(\boldsymbol{\omega}) \right]} = \frac{\mathbb{E} \left[ \sum_{m \in \mathcal{P}_c} g_m(\boldsymbol{\omega}) \right]}{\mathbb{E} \left[ \sum_{m \in \mathcal{P}_c} h_m(\boldsymbol{\omega}) \right]}.
\end{aligned}
$$

As a consequence, with $\boldsymbol{\beta}_c = \boldsymbol{\beta}_c^*$, the partition scheme must achieve lower weighted mean squared error than any globally shared $\widehat{\boldsymbol{\beta}}$, that is, $J(\boldsymbol{\beta}_1 = \boldsymbol{\beta}_1^*, \ldots, \boldsymbol{\beta}_C = \boldsymbol{\beta}_C^*) \leq J(\boldsymbol{\beta}_1 = \widehat{\boldsymbol{\beta}}, \ldots, \boldsymbol{\beta}_C = \widehat{\boldsymbol{\beta}})$.

In fact, with $\boldsymbol{\beta}_c = \boldsymbol{\beta}_c^*$, the partition scheme usually enjoys much lower error than adopting a globally shared coefficient. To see the error reduction of using the partitioned strategy, we first have

$$
\begin{aligned}
& J(\boldsymbol{\beta}_1 = \widehat{\boldsymbol{\beta}}, \dots, \boldsymbol{\beta}_C = \widehat{\boldsymbol{\beta}}) \\
& = \sum_{c=1}^{C} \sum_{m \in \mathcal{P}_c} \frac{\exp\left(\mathbf{q}_n^\top \mathbf{k}_m\right)}{\sum_{m' \in U} \exp\left(\mathbf{q}_n^\top \mathbf{k}_{m'}\right)} \left\| \widehat{\boldsymbol{\beta}} - \boldsymbol{\beta}_m^* \right\|^2 \\
& = \sum_{c=1}^{C} \sum_{m \in \mathcal{P}_c} \frac{\exp\left(\mathbf{q}_n^\top \mathbf{k}_m\right)}{\sum_{m' \in U} \exp\left(\mathbf{q}_n^\top \mathbf{k}_{m'}\right)} \left\| \widehat{\boldsymbol{\beta}} - \boldsymbol{\beta}_c + \boldsymbol{\beta}_c - \boldsymbol{\beta}_m^* \right\|^2 \\
& = \sum_{c=1}^{C} \sum_{m \in \mathcal{P}_c} \frac{\exp\left(\mathbf{q}_n^\top \mathbf{k}_m\right)}{\sum_{m' \in U} \exp\left(\mathbf{q}_n^\top \mathbf{k}_{m'}\right)} \left\| \widehat{\boldsymbol{\beta}} - \boldsymbol{\beta}_c + \boldsymbol{\beta}_c - \boldsymbol{\beta}_m^* \right\|^2 \\
& = \sum_{c=1}^{C} \sum_{m \in \mathcal{P}_c} \frac{\exp\left(\mathbf{q}_n^\top \mathbf{k}_m\right)}{\sum_{m' \in U} \exp\left(\mathbf{q}_n^\top \mathbf{k}_{m'}\right)} \left( \left\| \widehat{\boldsymbol{\beta}} - \boldsymbol{\beta}_c \right\|^2 + \left(\widehat{\boldsymbol{\beta}} - \boldsymbol{\beta}_c\right)^\top (\boldsymbol{\beta}_c - \boldsymbol{\beta}_m^*) + \|\boldsymbol{\beta}_c - \boldsymbol{\beta}_m^*\|^2 \right).
\end{aligned}
$$

Since

$$
\begin{aligned}
& \sum_{c=1}^{C} \sum_{m \in \mathcal{P}_c} \frac{\exp\left(\mathbf{q}_n^\top \mathbf{k}_m\right)}{\sum_{m' \in U} \exp\left(\mathbf{q}_n^\top \mathbf{k}_{m'}\right)} \left(\widehat{\boldsymbol{\beta}} - \boldsymbol{\beta}_c\right)^\top (\boldsymbol{\beta}_c - \boldsymbol{\beta}_m^*) \\
& = \sum_{c=1}^{C} \left(\widehat{\boldsymbol{\beta}} - \boldsymbol{\beta}_c\right)^\top \sum_{m \in \mathcal{P}_c} \frac{\exp\left(\mathbf{q}_n^\top \mathbf{k}_m\right)}{\sum_{m' \in U} \exp\left(\mathbf{q}_n^\top \mathbf{k}_{m'}\right)} (\boldsymbol{\beta}_c - \boldsymbol{\beta}_m^*) \\
& = \sum_{c=1}^{C} \left(\widehat{\boldsymbol{\beta}} - \boldsymbol{\beta}_c\right)^\top \sum_{m \in \mathcal{P}_c} \frac{\exp\left(\mathbf{q}_n^\top \mathbf{k}_m\right)}{\sum_{m' \in U} \exp\left(\mathbf{q}_n^\top \mathbf{k}_{m'}\right)} \left( \frac{\sum_{m \in \mathcal{P}_c} \exp\left(\mathbf{q}_n^\top \mathbf{k}_m\right) \mathbf{v}_m}{\sum_{m \in \mathcal{P}_c} \exp\left(\mathbf{q}_n^\top \mathbf{k}_m\right)} - \boldsymbol{\beta}_m^* \right) \\
& = \sum_{c=1}^{C} \left(\widehat{\boldsymbol{\beta}} - \boldsymbol{\beta}_c\right)^\top \frac{\sum_{m \in \mathcal{P}_c} \exp\left(\mathbf{q}_n^\top \mathbf{k}_m\right) (\mathbf{v}_m - \boldsymbol{\beta}_m^*)}{\sum_{m' \in U} \exp\left(\mathbf{q}_n^\top \mathbf{k}_{m'}\right)} \\
& = 0,
\end{aligned}
$$

plugging this result back we obtain

$$
\begin{aligned}
& J(\boldsymbol{\beta}_1 = \widehat{\boldsymbol{\beta}}, \dots, \boldsymbol{\beta}_C = \widehat{\boldsymbol{\beta}}) \\
& = \sum_{c=1}^{C} \sum_{m \in \mathcal{P}_c} \frac{\exp\left(\mathbf{q}_n^\top \mathbf{k}_m\right)}{\sum_{m' \in U} \exp\left(\mathbf{q}_n^\top \mathbf{k}_{m'}\right)} \left( \left\| \widehat{\boldsymbol{\beta}} - \boldsymbol{\beta}_c \right\|^2 + \left(\widehat{\boldsymbol{\beta}} - \boldsymbol{\beta}_c\right)^\top (\boldsymbol{\beta}_c - \boldsymbol{\beta}_m^*) + \|\boldsymbol{\beta}_c - \boldsymbol{\beta}_m^*\|^2 \right) \\
& = \sum_{c=1}^{C} \sum_{m \in \mathcal{P}_c} \frac{\exp\left(\mathbf{q}_n^\top \mathbf{k}_m\right)}{\sum_{m' \in U} \exp\left(\mathbf{q}_n^\top \mathbf{k}_{m'}\right)} \left( \left\| \widehat{\boldsymbol{\beta}} - \boldsymbol{\beta}_c \right\|^2 + \|\boldsymbol{\beta}_c - \boldsymbol{\beta}_m^*\|^2 \right) \\
& = \underbrace{\sum_{c=1}^{C} \frac{\sum_{m \in \mathcal{P}_c} \exp\left(\mathbf{q}_n^\top \mathbf{k}_m\right)}{\sum_{m' \in U} \exp\left(\mathbf{q}_n^\top \mathbf{k}_{m'}\right)} \left\| \widehat{\boldsymbol{\beta}} - \boldsymbol{\beta}_c \right\|^2}_{\geq 0} + \sum_{c=1}^{C} \sum_{m \in \mathcal{P}_c} \frac{\exp\left(\mathbf{q}_n^\top \mathbf{k}_m\right)}{\sum_{m' \in U} \exp\left(\mathbf{q}_n^\top \mathbf{k}_{m'}\right)} \|\boldsymbol{\beta}_c - \boldsymbol{\beta}_m^*\|^2 \\
& \geq \sum_{c=1}^{C} \sum_{m \in \mathcal{P}_c} \frac{\exp\left(\mathbf{q}_n^\top \mathbf{k}_m\right)}{\sum_{m' \in U} \exp\left(\mathbf{q}_n^\top \mathbf{k}_{m'}\right)} \|\boldsymbol{\beta}_c - \boldsymbol{\beta}_m^*\|^2.
\end{aligned}
$$

The last inequality holds since the first term is always non-negative. Note that the first term computes the squared error between $\widehat{\boldsymbol{\beta}}$ and each $\boldsymbol{\beta}_c$, weighted by the sum of attention scores over the corresponding subset. As a result, it is usually positive and the error reduction is significant if each $\boldsymbol{\beta}_c$ deviates from $\widehat{\boldsymbol{\beta}}$ a lot. However, although the optimal coefficient in the partitioning always leads to lower error to the optimal individual coefficient, note that it does not necessarily yield lower estimation variance. $\qquad \square$

Table 8: Classification results on `ImageNet1k` dataset under different hyper-parameter configurations of EVA. By default, we set $|E| = 49$ and $C = 49$ across all variants below.

| Component | Specification | Top-1 Acc. |
|---|---|---|
| Partition Scheme of $\{\mathcal{P}_1, \ldots, \mathcal{P}_C\}$ | partition over $[M] \setminus E$ | 76.53 |
| | partition over $[M]$ | 76.39 |
| Parameterization of $\sigma(\cdot)$ | $\sigma(\cdot) = \text{LN}(\text{Linear}(\cdot))$ | 76.67 |
| | $\sigma(\cdot) = \text{Identity}(\cdot)$ | 75.95 |
| Number of Groups ($C = 1$) | Number of Samples = 1 | 74.10 |
| | Number of Samples = 49 | 76.39 |
| Number of Groups ($C = 49$) | Number of Samples = 1 | 76.67 |
| | Number of Samples = 49 | 76.75 |
| Proposal Parameterization $q_c(\omega) := \mathcal{N}(\omega; \boldsymbol{\mu}_c, \mathbf{I})$ | $\boldsymbol{\mu}_c = \widetilde{\mathbf{q}}_c + \widetilde{\mathbf{k}}_c$ | 76.67 |
| | $\boldsymbol{\mu}_c = \widetilde{\mathbf{q}}_c$ | 76.77 |
| | $\boldsymbol{\mu}_c = \mathbf{0}$ | 76.24 |
| | $\boldsymbol{\mu}_c = \text{Trainable parameters}$ | 76.39 |
| Softmax | | 77.16 |

### B.7 DERIVATION OF EQUATIONS 14 AND 15

According to the definition of $g_m(\cdot)$ and $h_m(\cdot)$ in §3.1, for the vanilla RFA (Equation 4) we have

$$\text{RFA}(\mathbf{q}_n, \mathbf{K}, \mathbf{V}) = \frac{\sum_{s=1}^{S} \frac{p_n(\omega_s)}{q(\omega_s)} f(\omega_s)}{\sum_{s=1}^{S} \frac{p_n(\omega_s)}{q(\omega_s)}} = \frac{g(\boldsymbol{\omega})}{h(\boldsymbol{\omega})} = \frac{\sum_{m=1}^{M} g_m(\boldsymbol{\omega})}{\sum_{m=1}^{M} h_m(\boldsymbol{\omega})}.$$

Besides, since $\mathbf{v}_m = g_m(\boldsymbol{\omega})/h_m(\boldsymbol{\omega})$ and $\widehat{\boldsymbol{\beta}}_c(\boldsymbol{\omega}) = \left(\sum_{m \in \mathcal{P}_c} g_m(\boldsymbol{\omega})\right) / \left(\sum_{m \in \mathcal{P}_c} h_m(\boldsymbol{\omega})\right)$, we can re-write EVA as

$$\text{EVA}(\mathbf{q}_n, \mathbf{K}, \mathbf{V}) := \widetilde{g}(\boldsymbol{\omega})$$

$$= \sum_{m \in E} \widetilde{g}_m(\boldsymbol{\omega}) + \sum_{m \notin E} \widetilde{g}_m(\boldsymbol{\omega})$$

$$= \sum_{m \in E} \frac{\exp(\mathbf{q}_n^\top \mathbf{k}_m)}{Z} \mathbf{v}_m + \sum_{c=1}^{C} \frac{\sum_{m \in \mathcal{P}_c} \exp(\mathbf{q}_n^\top \mathbf{k}_m)}{Z} \widehat{\boldsymbol{\beta}}_c(\boldsymbol{\omega})$$

$$= \sum_{m \in E} \frac{\exp(\mathbf{q}_n^\top \mathbf{k}_m)}{Z} \frac{g_m(\boldsymbol{\omega})}{h_m(\boldsymbol{\omega})} + \sum_{c=1}^{C} \frac{\sum_{m \in \mathcal{P}_c} \exp(\mathbf{q}_n^\top \mathbf{k}_m)}{Z} \frac{\sum_{m \in \mathcal{P}_c} g_m(\boldsymbol{\omega})}{\sum_{m \in \mathcal{P}_c} h_m(\boldsymbol{\omega})}.$$

## C MORE IMPLEMENTATION DETAILS FOR EVA

In this section, we provide more details of EVA. We also conduct a comprehensive ablation study to test the effect of different components in our implementation and report the results in Table 8. The pseudo-code for EVA is listed in Algorithm 1.

**Approximating $\sum_{m \in \mathcal{P}_c} \exp(\mathbf{q}_n^\top \mathbf{k}_m)$ and Parameterizing $\widetilde{\mathbf{k}}_c$.** In our implementation, we approximate the sum of exponentials as $\sum_{m \in \mathcal{P}_c} \exp(\mathbf{q}_n^\top \mathbf{k}_m) \approx \exp(\mathbf{q}_n^\top \widetilde{\mathbf{k}}_c)$. Here we provide an informal justification for this approximation.

Our main motivation for such approximation is based on the simple intuition that the sum of exponentials grows as fast as the maximum exponential value, as reflected by the following inequality,

$$\max_{m \in \mathcal{P}_c} \exp(\mathbf{q}_n^\top \mathbf{k}_m) \leq \sum_{m \in \mathcal{P}_c} \exp(\mathbf{q}_n^\top \mathbf{k}_m) \leq |\mathcal{P}_c| \max_{m \in \mathcal{P}_c} \exp(\mathbf{q}_n^\top \mathbf{k}_m).$$

This means we can approximate the sum of exponentials by first computing the group representative $\widetilde{\mathbf{k}}_c := \arg\max_{\mathbf{k}_m \in \{\mathbf{k}_m | m \in \mathcal{P}_c\}} \exp(\mathbf{q}_n^\top \mathbf{k}_m)$, evaluating the corresponding exponential $\exp(\mathbf{q}_n^\top \widetilde{\mathbf{k}}_c)$

and then multiplying it by some scalar. Since computing the argmax operation still needs to compare each exponential dot-product, it will still incur quadratic computational costs. To circumvent this, we adopt a heuristic strategy that computes a learnable group representation, which attempts to compensate for the approximation error while only evaluating one exponential dot product.

Through preliminary experiments, we try various choices to compute the representative vector of each subset, such as max and average pooling; however, we found these strategies produce almost equally good performance. As a result, we adopt the average pooling by default due to its simplicity. To be specific, we implement it as

$$\widetilde{\mathbf{k}}_c = \sigma \left( \frac{1}{|\mathcal{P}_c|} \sum_{m \in \mathcal{P}_c} \mathbf{k}_m \right), \tag{26}$$

where $\sigma(\cdot)$ is a trainable linear projection with the same hidden dimension size as inputs, followed by a layer normalization operation (Ba et al., 2016) to stabilize training. We leave further improving the approximation, such as deriving tighter error bounds or using more expressive pooling methods (Zaheer et al., 2017; Ou et al., 2022) as future work.

**Parameterizing $\widehat{\boldsymbol{\beta}}_c(\boldsymbol{\omega})$.**    As discussed in §4.1, we have

$$\widehat{\boldsymbol{\beta}}_c(\boldsymbol{\omega}) = \frac{\sum_{m \in \mathcal{P}_c} g_m(\boldsymbol{\omega})}{\sum_{m \in \mathcal{P}_c} h_m(\boldsymbol{\omega})} = \frac{\sum_{s=1}^{S} \frac{\mathcal{N}(\omega_s; 0, \mathbf{I})}{Z q(\omega_s)} \sum_{m \in \mathcal{P}_c} \xi(\mathbf{q}_n, \omega_s) \xi(\mathbf{k}_m, \omega_s) \mathbf{v}_m}{\sum_{s=1}^{S} \frac{\mathcal{N}(\omega_s; 0, \mathbf{I})}{Z q(\omega_s)} \sum_{m \in \mathcal{P}_c} \xi(\mathbf{q}_n, \omega_s) \xi(\mathbf{k}_m, \omega_s)}.$$

Compared to the SNIS formulation of vanilla RFA Equation 4, we can express it as

$$\mathsf{RFA}(\mathbf{q}_n, \mathbf{K}, \mathbf{V}) = \frac{\sum_{s=1}^{S} \frac{p_n(\omega_s)}{q(\omega_s)} f(\omega_s)}{\sum_{s=1}^{S} \frac{p_n(\omega_s)}{q(\omega_s)}} = \frac{\sum_{m=1}^{M} g_m(\boldsymbol{\omega})}{\sum_{m=1}^{M} h_m(\boldsymbol{\omega})}.$$

We can think of each coefficient $\widehat{\boldsymbol{\beta}}_c(\boldsymbol{\omega})$ as computing the output of a localized RFA for each group $\mathcal{P}_c$. From this perspective, we can recast each coefficient $\widehat{\boldsymbol{\beta}}_c(\boldsymbol{\omega})$ as an SNIS estimator as well, which tries to estimate

$$\mathbb{E}_{\omega \sim p_c(\omega)} \left[ f_c(\omega) \right] = \sum_{m \in \mathcal{P}_c} \frac{\exp \left( \mathbf{q}_n^\top \mathbf{k}_m \right)}{\sum_{m' \in \mathcal{P}_c} \exp \left( \mathbf{q}_n^\top \mathbf{k}_{m'} \right)} \mathbf{v}_m \tag{27}$$

where

$$f_c(\omega) := \frac{\sum_{m \in \mathcal{P}_c} \xi(\mathbf{q}_n, \omega) \xi(\mathbf{k}_m, \omega) \mathbf{v}_m}{\sum_{m' \in \mathcal{P}_c} \xi(\mathbf{q}_n, \omega) \xi(\mathbf{k}_{m'}, \omega)},$$

$$p_c(\omega) := \frac{\mathcal{N}(\omega; 0, \mathbf{I}) \sum_{m \in \mathcal{P}_c} \xi(\mathbf{q}_n, \omega)^\top \xi(\mathbf{k}_m, \omega)}{\sum_{m' \in \mathcal{P}_c} \exp \left( \mathbf{q}_n^\top \mathbf{k}_{m'} \right)}$$

$$= \sum_{m \in \mathcal{P}_c} \frac{\exp \left( \mathbf{q}_n^\top \mathbf{k}_m \right)}{\sum_{m' \in \mathcal{P}_c} \exp \left( \mathbf{q}_n^\top \mathbf{k}_{m'} \right)} \mathcal{N}(\omega; \mathbf{q}_n + \mathbf{k}_m, \mathbf{I}).$$

This interpretation indicates that a good proposal distribution $q_c(\omega)$ should be specific to each subset $\mathcal{P}_c$. To get close to the true distribution $p_c(\omega)$ while keeping efficient computation, Zheng et al. (2022b) suggests parameterizing the proposal distribution as

$$q_c(\omega) := \mathcal{N}(\omega; \mu_c, \mathbf{I}) = \mathcal{N}(\omega; \widetilde{\mathbf{q}}_c + \widetilde{\mathbf{k}}_c, \mathbf{I}), \tag{28}$$

where $\widetilde{\mathbf{q}}_c$ is calculated similarly to Equation 26. We refer readers to Zheng et al. (2022b) for more discussions about the parameterization choice of proposal distributions. We conduct further ablation studies to test the effect of proposal parameterizations in our proposed model, as shown in Table 8. In particular, we found our model is robust to different parameterization approaches.

The essence in making the algorithm memory-efficient is to use only *one* sample in calculating $\widehat{\boldsymbol{\beta}}_c(\boldsymbol{\omega})$. In this case, we have

$$
\begin{aligned}
\widehat{\boldsymbol{\beta}}_c(\boldsymbol{\omega}) &= \frac{\sum_{m\in\mathcal{P}_c} g_m(\boldsymbol{\omega})}{\sum_{m\in\mathcal{P}_c} h_m(\boldsymbol{\omega})} \\
&= \frac{\frac{\mathcal{N}(\omega^c;0,\mathbf{I})}{Zq_c(\omega^c)}\sum_{m\in\mathcal{P}_c}\xi(\mathbf{q}_n,\omega^c)\xi(\mathbf{k}_m,\omega^c)\mathbf{v}_m}{\frac{\mathcal{N}(\omega^c;0,\mathbf{I})}{Zq_c(\omega^c)}\sum_{m\in\mathcal{P}_c}\xi(\mathbf{q}_n,\omega^c)\xi(\mathbf{k}_m,\omega^c)} \\
&= \frac{\frac{\mathcal{N}(\omega^c;0,\mathbf{I})}{Zq_c(\omega^c)}\xi(\mathbf{q}_n,\omega^c)\sum_{m\in\mathcal{P}_c}\xi(\mathbf{k}_m,\omega^c)\mathbf{v}_m}{\frac{\mathcal{N}(\omega^c;0,\mathbf{I})}{Zq_c(\omega^c)}\xi(\mathbf{q}_n,\omega^c)\sum_{m\in\mathcal{P}_c}\xi(\mathbf{k}_m,\omega^c)} \\
&= \frac{\sum_{m\in\mathcal{P}_c}\xi(\mathbf{k}_m,\omega^c)\mathbf{v}_m}{\sum_{m\in\mathcal{P}_c}\xi(\mathbf{k}_m,\omega^c)}, \qquad w^c\sim q_c(\omega).
\end{aligned}
$$

Since this degenerated formulation eliminates the dependence on individual queries $\mathbf{q}_n$, we could pre-compute $\widehat{\boldsymbol{\beta}}_c(\boldsymbol{\omega})$ for each $\mathcal{P}_c$, and then re-uses them for each query, which takes up $\mathcal{O}(Cd)$ memory. If multiple samples are used instead, the influence of queries needs to be explicitly taken into account and thus we need to compute a distinct $\widehat{\boldsymbol{\beta}}_c(\boldsymbol{\omega})$ for each query, leading to $\mathcal{O}(NCd)$ memory usage, which incurs a significant compute overhead.

On the other hand, if we set $C = 1$, that is, using a shared $\widehat{\boldsymbol{\beta}}_c(\boldsymbol{\omega})$ over all $m \notin E$, our approach does not suffer from this issue, since the memory usage is at most $\mathcal{O}(Nd)$. To investigate the effect of using larger $C$ or increasing the number of samples, we conduct an ablative analysis as in Table 8, and find that 1) when $C = 1$, the performance degrades a lot when using one sample, which can be largely improved by adopting more samples; while when $C > 1$, our partitioning strategy dominates and increasing the number of samples only improves performance marginally. This also validates the effectiveness of adopting a finer-grained treatment over control variates.

**Partitioning Strategy.** EVA significantly improves random feature approximation by trying to locally estimate each subset of tokens, which is a much easier task than approximating the whole sequence as in previous RFA methods. To achieve this, EVA partitions the whole token sequence into multiple subsets according to the current query position $n$, which is denoted by $\{E^n, \mathcal{P}_1^n, \mathcal{P}_2^n, \ldots, \mathcal{P}_C^n\}_{n=1}^N$.[3] For elements in subset $E^n$, we optimize the control variate coefficient to give an exact estimate for each single token $m \in E^n$. In addition, we impose T5-style relative positional encoding (Raffel et al., 2020a) over elements in $E^n$. While for some other subset $\mathcal{P}_c$, we employ the shared coefficient to approximate all tokens belonging to $\mathcal{P}_c$. We assume all $E^1, \ldots, E^N$ are of the same cardinality $K$, and $|\mathcal{P}_c^n|$ is the same for any $c = 1, \ldots, C$ and $n = 1, \ldots, N$.

The partition strategy $\{E^n, \mathcal{P}_1^n, \mathcal{P}_2^n, \ldots, \mathcal{P}_C^n\}_{n=1}^N$ is decided based on a simple criterion:

- for $E^n$, it contains $K$ local neighbors with respect to each query $n$. To further simplify implementation and reduce memory usage, we chunk the whole sequence into contiguous blocks of size $K$, and all adjacent queries belonging to the same block will share this block as the subset $E^n$;
- as for $\mathcal{P}_1^n, \mathcal{P}_2^n, \ldots, \mathcal{P}_C^n$, we follow a similar treatment by splitting the complement $[M] \setminus E^n$ into $C$ contiguous chunks of the same size. For ease of implementation, we simply partition the whole index set $[M]$ into multiple groups instead of $[M] \setminus E^n$, which circumvents the overload for explicitly performing set difference operations in practical implementation. Although this leads to extra approximation error, this amounts to putting more attention weights on tokens belonging to the subset $E$ and we found this approximation does not lead to performance degradation (Table 8).

## D  A CAUSAL VARIANT OF EVA

In this section, we describe the causal variant of EVA, where each query can only attend to historical tokens. Thanks to the partitioning scheme, all future information with respect to the current query token can be masked conveniently. Following the formulation of EVA, we partition the whole sequence into $C + 1$ subsets $\{E^n, \mathcal{P}_1^n, \mathcal{P}_2^n, \ldots, \mathcal{P}_C^n\}$ with respect to each query $\mathbf{q}_n$. To fulfill the

---

[3]Here we add the superscript $n$ to reflect the dependence on query position $n$.

---

**Algorithm 1** Pseudo-code for EVA

---

**Input:** the randomized mapping $\xi(\cdot, \cdot)$, queries $\mathbf{Q} := \{\mathbf{q}_n\}_{n=1}^N$, keys $\mathbf{K} := \{\mathbf{k}_m\}_{m=1}^M$, values $\mathbf{V} := \{\mathbf{v}_m\}_{m=1}^M$ and partitions of the sequence $\{E^n, \mathcal{P}_1^n, \mathcal{P}_2^n, \ldots, \mathcal{P}_C^n\}_{n=1}^N$;
**Output:** attention output $\mathbf{Y} := \{\mathbf{y}_n\}_{n=1}^N$;

 

**for** $c = 1, 2, \ldots, C$ **do**
    Compute $\widetilde{\mathbf{k}}_c$ according to Equation 26;

    Compute $q_c(\omega)$ according to Equation 28;
    Sample $\omega_c \sim q_c(\omega)$;                    $\triangleright$ During inference, simply set $\omega_c = \mathbb{E}_{q_c(\omega)}[\omega]$
    Compute $\widehat{\boldsymbol{\beta}}_c(\boldsymbol{\omega}) = \sum_{m \in \mathcal{P}_c^n} \frac{\xi(\mathbf{k}_m, \omega_c)}{\sum_{m \in \mathcal{P}_c^n} \xi(\mathbf{k}_m, \omega_c)} \mathbf{v}_m$;
**end for**
**for** $n = 1, 2, \ldots, N$ **do**
    Compute $\mathcal{S} = \sum_{m \in E^n} \exp\left(\mathbf{q}_n^\top \mathbf{k}_m\right) \mathbf{v}_m$; $\triangleright$ Compute attention scores in the selected subset $E$
    Compute $\mathcal{R} = \sum_{c=1}^C \exp\left(\mathbf{q}_n^\top \widetilde{\mathbf{k}}_c\right) \widehat{\boldsymbol{\beta}}_c(\boldsymbol{\omega})$;       $\triangleright$ Compute approx. expected control variates
    Compute $Z = \sum_{m \in E^n} \exp\left(\mathbf{q}_n^\top \mathbf{k}_m\right) + \sum_{c=1}^C \exp\left(\mathbf{q}_n^\top \widetilde{\mathbf{k}}_c\right)$;
    Compute $\mathbf{y}_n = (\mathcal{S} + \mathcal{R})/Z$;
**end for**
**Return** $\mathbf{Y} := [\mathbf{y}_1, \ldots, \mathbf{y}_N]$.

---

causal requirement, we design two different types of masking matrices to deal with both $E^n$ and $\{\mathcal{P}_c^n\}_{c=1}^C$ respectively.

- For $E^n$, we adopt a single lower-triangular matrix with shape $K \times K$ (recall that each set $E^n$ is of size $K$) to mask future tokens locally, similar to the case of standard decoder softmax attention. Future tokens that do not belong to $E^n$ are handled by masking functions for $\{\mathcal{P}_c^n\}_{c=1}^C$, as described below.
- For $\{\mathcal{P}_c^n\}_{c=1}^C$, we make use of the fact $n \in E^n$. Since any $\mathcal{P}_c^n$ and $E^n$ are disjoint, we only need to mask all subsets $\mathcal{P}_c^n$ that appear after $E^n$. This amounts to first allocating a lower-triangular matrix with shape $C \times C$, and then conducting future masking at a subset level.

The pseudo-code for the causal variant of EVA is listed in Algorithm 2.

# E  EXPERIMENTAL DETAILS

All of our experiments are conducted with at most 16 NVIDIA V100 GPUs.

## E.1  EFFICIENT ATTENTION BASELINES

We compare our proposed attention mechanism EVA against various baselines:

- Performer (Choromanski et al., 2021), which uses the plain random features to approximate softmax attention;
- LARA (Zheng et al., 2022b), an advanced RF approximation that makes use of multiple adaptive proposals to construct the SNIS estimator;
- Linformer (Wang et al., 2020), a low-rank approximation that uses a learnable matrix to project the key-value sequence into a shorter one;
- Nyströmformer (Xiong et al., 2021b), a low-rank approximation that adopts the Nyström method to approximate softmax attention map with a sub-sampled matrix;
- Local attention (Child et al., 2019), a simple sparse approximation that splits the whole sequence into multiple blocks and only allows the query to attend to tokens within the same block;
- Reformer (Kitaev et al., 2020), a sparse approximation where hash functions are used to adaptively distribute sequence tokens into multiple buckets, and each token can only attend to tokens within the same bucket;

---

**Algorithm 2** Pseudo-code for Causal EVA

---

**Input:** the randomized mapping $\xi(\cdot,\cdot)$, queries $\mathbf{Q} \coloneqq \{\mathbf{q}_n\}_{n=1}^N$, keys $\mathbf{K} \coloneqq \{\mathbf{k}_m\}_{m=1}^M$, values $\mathbf{V} \coloneqq \{\mathbf{v}_m\}_{m=1}^M$, and partitions of the sequence $\{E^n, \mathcal{P}_1^n, \mathcal{P}_2^n, \ldots, \mathcal{P}_C^n\}_{n=1}^N$;
**Output:** attention output $\mathbf{Y} \coloneqq \{\mathbf{y}_n\}_{n=1}^N$;

**for** $c = 1, 2, \ldots, C$ **do**
    Compute $\widetilde{\mathbf{k}}_c$ according to Equation 26;

    Compute $q_c(\omega)$ according to Equation 28;
    Sample $\omega_c \sim q_c(\omega)$;            ▷ During inference, simply set $\omega_c = \mathbb{E}_{q_c(\omega)}[\omega]$
    Compute $\widehat{\boldsymbol{\beta}}_c(\boldsymbol{\omega}) = \sum_{m \in \mathcal{P}_c^n} \frac{\xi(\mathbf{k}_m, \omega_c)}{\sum_{m \in \mathcal{P}_c^n} \xi(\mathbf{k}_m, \omega_c)} \mathbf{v}_m$;
**end for**
Let $K \leftarrow |E^N|$;            ▷ we assume all $E^n$ are the same in size
Initialize $\mathbf{M}^E \in \{0,1\}^{K \times K}$ such that $\mathbf{M}_{i,j}^E = \mathbb{1}_{i \leq j}$;            ▷ Intra-$E$ masking matrix
Initialize $\mathbf{M}^{\mathcal{P}} \in \{0,1\}^{C \times C}$ such that $\mathbf{M}_{c,t}^{\mathcal{P}} = \mathbb{1}_{c \leq t}$;            ▷ Inter-$\mathcal{P}$ masking matrix
**for** $n = 1, 2, \ldots, N$ **do**
    Find index $t$ such that $\mathcal{P}_t^n$ is the most recent chunk on the left of $E$;
    Let $b^n \leftarrow \min_i \{i : i \in E^n\}$;    ▷ The least position within $E^n$; used for shifting token indices.

    ▷ The same masking matrix $\mathbf{M}^E$ can be reused across $n$ via shifting token positions by $b^n$.
    Compute $\mathcal{S} = \sum_{m \in E^n} \mathbf{M}_{m-b^n, n-b^n}^E \exp\left(\mathbf{q}_n^\top \mathbf{k}_m\right) \mathbf{v}_m$;
    Compute $\mathcal{R} = \sum_{c=1}^C \mathbf{M}_{c,t}^{\mathcal{P}} \exp\left(\mathbf{q}_n^\top \widetilde{\mathbf{k}}_c\right) \widehat{\boldsymbol{\beta}}_c(\boldsymbol{\omega})$;
    Compute $Z = \sum_{m \in E^n} \mathbf{M}_{m-b^n, n-b^n}^E \exp\left(\mathbf{q}_n^\top \mathbf{k}_m\right) + \sum_{c=1}^C \mathbf{M}_{c,t}^{\mathcal{P}} \exp\left(\mathbf{q}_n^\top \widetilde{\mathbf{k}}_c\right)$;
    Compute $\mathbf{y}_n = (\mathcal{S} + \mathcal{R})/Z$;
**end for**
**Return** $\mathbf{Y} \coloneqq [\mathbf{y}_1, \ldots, \mathbf{y}_N]$.

---

- Scatterbrain (Chen et al., 2021a), an approach that combines Performer and sparse attention. The details can be found in Appendix G. Here we implement the sparse module as a simple local attention to ensure a fair comparison;
- Combiner (Ren et al., 2021), a probabilistic approach that constructs a structured factorization over the softmax probability distribution via a sparse mechanism. Combiner allows both direct and indirect calculations of conditional probabilities, where the direct probability is implemented as the sparse mechanism while the indirect probability is implemented through a local abstraction over a group of tokens. Similarly, we implement the sparse mechanism as a simple local attention, which corresponds to the Combiner-Fixed variant (Ren et al., 2021);
- Transformer-LS, or Long-Short (Zhu et al., 2021), which is proposed to model long-term and short-term dependencies via low-rank structures and local attention respectively. The low-rank structure is defined as an input-dependent weight matrix that compresses the sequence into a shorter one; while the local attention is defined similarly as above.

Note that for all mechanisms that involve a local attention, we split the sequence into *non-overlapping* blocks (or 2D windows in terms of images) and each query can only attend to tokens within the same block. We also use the relative positional embedding (Raffel et al., 2020b; Liu et al., 2021) within the local attention computation. Unlike Transformer-LS (Zhu et al., 2021) that allows each query to attend to multiple blocks, we do not use this extension as we find greatly increases memory consumption, although it does improve the model performance.

### E.2 IMAGE CLASSIFICATION

Through the experiments on image classification, we consider four different vision transformer (ViT) architectures:

Table 9: Our hyper-parameter configuration for different attention mechanisms on DeiT-Tiny-784.

| Attention | Hyper-parameter configuration on image classification | |
|---|---|---|
| Local attention | Window size | 49 |
| Scatterbrain (Kitaev et al., 2020) | umber of random feature samples | 96 |
| | Local attention window size | 49 |
| Nyströmformer (Xiong et al., 2021b) | Number of landmarks | 49 |
| Performer (Choromanski et al., 2021) | Number of random feature samples | 128 |
| | Type of random feature | Positive |
| Combiner (Ren et al., 2021) | Mode | Fixed |
| | Span size | 49 |
| | Conditional distribution parameterization | DeepSets-Max |
| Transformer-LS (Zhu et al., 2021) | Dynamic projection dimension | 16 |
| | Local window size | 49 |
| EVA (Ours) | Number of partitioned groups ($C$) | 49 |
| | Size of $E$ | 49 |

- DeiT-Tiny (Touvron et al., 2021), which maintains the sequence length as 196 across all transformer layers. For the particular tiny variant, the number of transformer layers is set to 12, the embedding dimension is set to 196 and the number of heads is 3;
- DeiT-Small (Touvron et al., 2021), which scales the embedding dimension and number of attention heads in DeiT-Tiny up to 384 and 6, respectively;
- DeiT-Tiny-784, where the architecture is the same as DeiT-Tiny but the patch size in the tokenization step is decreased from 16 to 8. This effectively increases the sequence length from 196 to 784, which we found consistently improves predictive accuracy at the cost of significantly increased time and memory consumption. Under this setting, we also see clearer differences among these attention variants and it helps better evaluate the ability of different attention models to learn visual representations;
- PVT-v2-B3 (Wang et al., 2021b), a pyramidal transformer architecture that processes much longer token sequences at early layers and progressively reduces the sequence length to form a hierarchical structure. It patchifies input images into 3136 ($56 \times 56$) tokens, and then processes the sequence through 4 stages. Each stage contains several transformer layers and a down-sampling operation, which reduces the sequence length by a factor of 4 and increases the embedding dimension by $2\times$. Due to the prohibitively long sequences initially, PVT applies an additional down-sampling module on input sequences to obtain key and value vectors, which are then passed through a normal softmax attention mechanism. To evaluate different RF approximations, we remove the down-sampling operation and directly operate on the original sequence length, which results in much fewer model parameters than vanilla PVT-v2-B3. We refer readers to Wang et al. (2021b) for detailed architecture configurations.

For training, we do not use the [CLS] token for classification (Touvron et al., 2021); instead, we pool over the output of the last transformer layer to extract features and feed them into the classifier head. We followed the same protocol to train all model variants. Closely following DeiT Touvron et al. (2021), we employ the AdamW (Loshchilov & Hutter, 2019) optimizer to train models for 300 epochs, where the number of warm-up epochs is 10, the learning rate is 0.001 with cosine learning rate decay (Loshchilov & Hutter, 2016), and batch size is set to 1024. The adopted augmentation and regularization are the same as DeiT, except that we remove repeated augmentation (Hoffer et al., 2020) in DeiT models as it often slows down convergence, as also observed in previous studies (Xiao et al., 2021).[4] The specific configurations of each attention mechanism on DeiT-Tiny-784 are listed in Table 9. The hyper-parameter setup for each attention variant follows previous practices (Wang et al., 2021a;b; Zheng et al., 2022b) closely to ensure a similar computational cost.

**Comparison to State-of-the-Art Model Architectures.** We also compare our model against recent state-of-the-art (SOTA) model architectures with similar parameter sizes on ImageNet1k benchmark. As reported in Table 10, we observe that PVT-v2 (Wang et al., 2021b) with EVA greatly

---

[4]we retain the repeated augmentation technique in training PVT to be consistent with the original training protocol in Wang et al. (2021b).

Table 10: Results on `ImageNet1k` dataset compared with SOTA model architectures.

| Model | # Param. | FLOPs | Top-1 Acc. |
|---|---|---|---|
| PVT-v1-M (Wang et al., 2021a) | 44M | 6.7G | 81.2 |
| RegNetY-8G (Radosavovic et al., 2020) | 39M | 8.0G | 81.7 |
| CvT-21 (Wu et al., 2021) | 32M | 7.1G | 82.5 |
| SOFT-M (Lu et al., 2021) | 45M | 7.2G | 82.9 |
| RegNetY-16G (Radosavovic et al., 2020) | 84M | 16.0G | 82.9 |
| UniFormer-S (Li et al., 2022) | 22M | 3.6G | 82.9 |
| Swin-S (Liu et al., 2021) | 50M | 8.7G | 83.0 |
| Swin-B (Liu et al., 2021) | 88M | 15.4G | 83.3 |
| RegionViT-M (Chen et al., 2021b) | 42M | 7.9G | 83.4 |
| ViL-M (Zhang et al., 2021) | 40M | 9.1G | 83.5 |
| Focal-S (Yang et al., 2021) | 51M | 9.1G | 83.5 |
| PVT-v2-b3 + LARA (Zheng et al., 2022b) | 40M | 7.7G | 83.6 |
| MaxViT-T (Tu et al., 2022) | 31M | 5.6G | 83.6 |
| UniFormer-B (Li et al., 2022) | 50M | 8.3G | 83.9 |
| PVT-v2-b3 (Wang et al., 2021b) | 45M | 6.9G | 83.1 |
| PVT-v2-b3 + EVA | 36M | 7.4G | 83.7 |

improves the predictive accuracy and performs competitively with recent SOTA architectures while using fewer parameters and FLOPs.

### E.3  MACHINE TRANSLATION AND LANGUAGE MODELING

Our implementation for all language tasks is based on FairSeq toolkit (Ott et al., 2019). To compare different methods, we report BLEU scores on the test set as the main metric for MT and perplexity for both Autoregressive LM and MLM tasks. For the hyper-parameters $|E|$ and $C$ in EVA, we set $|E| = 2C$ by default, as we find that this choice attains a good trade-off between performance and computational costs across various tasks; while for $C$, it is determined based on previous practice for each task. Here we provide the detailed experimental protocol for each task.

**Masked Language Modeling.**  Following the standard pretraining practice as in RoBERTa (Liu et al., 2019), in MLM, we aim to reconstruct a subset of tokens in the input sequence that are randomly masked out, which is the core element of BERT-style natural language pretraining (Devlin et al., 2019). This setting allows us to investigate the generalization ability of our model on larger model sizes and much more data. The task performance is measured with validation perplexity, which reflects how well the model fits the pretraining corpus and also exhibits good correlations with downstream task metrics. For the used corpus `Books3`, we randomly select 100 books without replacement for the validation split, similar to the setup in C4 dataset (Raffel et al., 2020b). For the model, we use the RoBERTa-base architecture (Liu et al., 2019), where all the layer normalization operations (Ba et al., 2016) are placed before attention and FFN blocks (i.e., we adopt the pre-norm architecture), which leads to much more stable training for efficient attention mechanisms. We replace all softmax attention with EVA to test its effectiveness. The training setting and attention-specific parameters, which follow previous studies (Xiong et al., 2021a) to ensure a similar computational cost, can be found in Table 11 and Table 12 respectively.

**Machine Translation.**  We follow Ott et al. (2018) to process `WMT14 En-De` dataset, resulting in around 4.5M/3K/3K English-German sentence pairs for training/validation/testing splits, respectively, and a shared vocabulary is obtained between the source and target language of around 32K BPE types. The architecture and training specifics closely follow Vaswani et al. (2017), as listed in Table 13. We follow the previous protocol Zheng et al. (2022b) by replacing all encoder self-attention blocks in the encoder-decoder Transformer with EVA. For EVA, we find it beneficial to introduce an overlapping variant of $E$, where we allow $E$ to be overlapped with each other. Following previous practice in the context of local attention (Xiong et al., 2021a), $E$ not only contains all elements within the designated chunk but also additionally includes half the tokens in its *neighboring* chunks. As a result, EVA-32 corresponds to $|E| = 32$ with a contiguous chunk size of 16. During inference, we follow the same setup as Zheng et al. (2022b) and average the last 10 model checkpoints to obtain the final model parameters. We apply beam search with size 4, length penalty 0.6, and compound split

Table 11: Our hyper-parameter configuration for Masked Language Modeling (MLM).

| Hyper-parameter | MLM |
|---|---|
| Number of transformer encoder layers | 12 |
| Hidden size | 768 |
| hidden size in FFN | 3072 |
| Number of attention heads | 12 |
| Batch size | 256 |
| Sequence length | $\{2048, 4096\}$ |
| Number of training steps | 200K |
| Number of warm-up steps | 5K |
| Weight decay rate | 0.01 |
| Peak Learning Rate | 1e-4 |
| Learning rate decay | Linear |
| Optimizer | Adam |
| Adam $\epsilon$ | 1e-6 |
| Adam $(\beta_1, \beta_2)$ | (0.9, 0.98) |
| Gradient Clipping | 0.0 |
| Dropout | 0.1 |
| Attention dropout (if applicable) | 0.0 |

Table 12: Our hyper-parameter configuration for different attention mechanisms on MLM task. * We used the exact positive random feature map (Choromanski et al., 2021) in our preliminary experiments. However, it failed to converge and exhibited substantial training instability. Therefore, we replace the positive random feature with a simple ReLU kernel function for MLM experiments, which yields better training performance.

| Attention | Hyper-parameter configuration on MLM | |
|---|---|---|
| Local attention | Window size | 256 |
| Linformer (Wang et al., 2020) | Projected dimension | 256 |
| Reformer (Kitaev et al., 2020) | Number of hashes | 4 |
| | Chunk size | 64 |
| Performer (Choromanski et al., 2021) | Number of random feature samples | 256 |
| | Type of random feature | ReLU[*] |
| LARA (Zheng et al., 2022b) | Number of landmarks | 256 |
| Combiner (Ren et al., 2021) | Mode | Fixed |
| | Span size | 256 |
| | Conditional distribution parameterization | DeepSets-Max |
| Transformer-LS (Zhu et al., 2021) | Dynamic projection dimension | 128 |
| | Local window size | 256 |
| EVA (Ours) | Number of partitioned groups ($C$) | 128 |
| | Size of $E$ | 256 |

post-processing. Since the input sequences in `WMT14 En-De` benchmark are much shorter than the other tasks considered in this paper (with an average sequence length of around 25 tokens), we start with $C = 8$, $|E| = 16$ and gradually increase $|E|$ to test the translation performance, similar to the setup in Ma et al. (2021); Zheng et al. (2022b). Note that increasing $C$ also leads to better translation quality, although we found the performance gain is slightly less effective than that of increasing $|E|$ (c.f. Tables 6 and 7).

**Autoregressive Language Modeling.** We consider `Wikitext-103` benchmark in this task, which consists of around 103M/218K/246K tokens for training/validation/testing splits, respectively. We adopt the vanilla transformer decoder architecture (Vaswani et al., 2017), replace all decoder self-attention modules in the Transformer with the causal EVA mechanism, and evaluate EVA under two different setups: 1) a standard 16-layer Transformer LM (with model sizes of around 247M) as in Baevski & Auli (2019), and 2) a larger 32-layer Transformer LM (with model sizes of around 450M) as in Kasai et al. (2021). We follow their hyper-parameter settings to train all models, where

Table 13: Our hyper-parameter configuration for machine translation.

| Hyper-parameter | Machine Translation |
|---|---|
| Number of transformer encoder layers | 6 |
| Number of transformer decoder layers | 6 |
| Hidden size | 512 |
| hidden size in FFN | 2048 |
| Number of attention heads | 8 |
| Maximum number of tokens in a batch | 32768 |
| Number of training steps | 300K |
| Number of warm-up steps | 6K |
| Weight decay rate | 0.0 |
| Peak Learning Rate | 0.0007 |
| Label Smoothing | 0.1 |
| Learning rate decay | Inverse square root |
| Optimizer | Adam |
| Adam $\epsilon$ | 1e-6 |
| Adam $(\beta_1, \beta_2)$ | (0.9, 0.98) |
| Gradient Clipping | 5.0 |
| Dropout | 0.1 |
| Attention dropout (if applicable) | 0.1 |

Table 14: Our hyper-parameter configuration for autoregressive language modeling.

| Hyper-parameter | LM in Baevski & Auli (2019) | LM in Kasai et al. (2021) |
|---|---|---|
| Number of transformer decoder layers | 16 | 32 |
| Hidden size | 1024 | 1024 |
| hidden size in FFN | 4096 | 4096 |
| Number of attention heads | 8 | 8 |
| Number of tokens in a batch | 65536 | 65536 |
| Number of training steps | 286K | 286K |
| Number of warm-up steps | 16K | 16K |
| Weight decay rate | 0.0 | 0.0 |
| Peak Learning Rate | 1.0 | 1.0 |
| Learning rate decay | cosine | cosine |
| Optimizer | nag | nag |
| Gradient Clipping | 0.1 | 0.1 |
| Dropout | 0.3 | 0.3 |
| LayerDrop | – | 0.2 |
| Attention dropout | 0.1 | 0.1 |

the corresponding configurations are listed in Table 14. [5] The vocabulary size is 267,744 with adaptive input embeddings (Baevski & Auli, 2019). During training, we set the sequence length to 512 and evaluate the validation/test PPL with various context window sizes in $\{256, 480\}$, aligning with previous work (Baevski & Auli, 2019; Kasai et al., 2021). For other random feature baselines, unfortunately, we failed to fully replicate their results as reported in Kasai et al. (2021), where RFA in our implementation achieved a test perplexity of 29.0 even under a 449M Transformer model. For EVA, we set $|E| = 128$ and $C = 64$ by default for both 16-layer and 32-layer settings, ensuring similar computational cost to previous work that also evaluates random feature methods (typically with 128 or 256 random-feature dimension size) on `Wikitext-103` language modeling task (Schlag et al., 2021; Kasai et al., 2021).

### E.4 EXPERIMENTAL SETTINGS OF EFFICIENCY COMPARISON

For the simulation experiment conducted in §5.3, we adopt the same transformer architecture across all attention variants. In particular, it uses 8 transformer layers, 192 embedding dimensions, and 2 attention heads so that longer sequences can fit into our devices. The batch size is set to 64 across

---

[5]The setup in Baevski & Auli (2019) can be found in the corresponding Fairseq training script: `https://github.com/pytorch/fairseq/blob/master/examples/language_model/README.adaptive_inputs.md`.

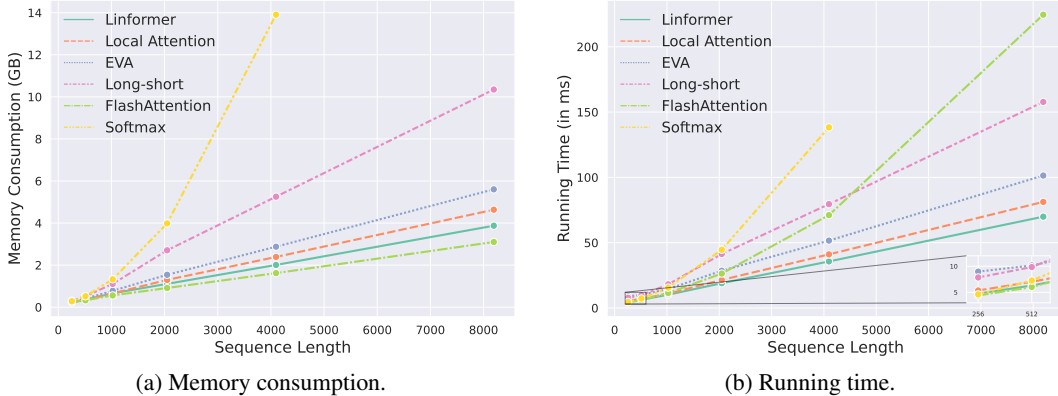

(a) Memory consumption.

(b) Running time.

Figure 2: **Left** and **right**: Additional empirical memory consumption and running time comparison for different attention mechanisms under various sequence lengths.

8 V100 GPUs, and the statistics are computed by averaging the results of 30 runs. Besides, in our ablation study, the efficiency metrics reported in Table 6 and Table 7 are evaluated under the same setup used during training.

**Remark on Modeling Short Sequences.** Unfortunately, similar to most previous efficient attention baselines, EVA also runs slower than softmax attention under shorter sequences (e.g., length of 128 or 256), but it soon catches up in running speed, and the reduction of memory consumption is still significant. Besides, in short-sequence settings (such as the case of DeiT-Tiny/Small with sequences of 196 tokens), EVA often performs on par with or better than conventional softmax attention (see Table 1), whereas most previous attention variants usually perform much worse. This implies EVA can achieve a better trade-off between efficiency and quality: for short sequences, EVA is possible to achieve stronger performance competitive with softmax attention (despite in longer running time); while for long sequences, EVA can be run much faster with less memory.

**Comparison to Memory-efficient Attention Mechanisms.** In this section, we conduct an empirical efficiency comparison between efficient approximate attention methods and FlashAttention, one of the memory-efficient attention mechanisms (Rabe & Staats, 2021; Dao et al., 2022) with optimized memory accesses. FlashAttention computes the exact softmax attention in an online manner without materializing the full attention matrix, achieving linear memory complexity with respect to sequence lengths; besides, both runtime and memory usage are further improved by minimizing IO accesses. We benchmark different attention modules on one NVIDIA GeForce RTX 3090 GPU, where we measure the memory usage and runtime of running a single attention block, consisting of 8 attention heads with 512 embedding dimension size, for both a forward and backward pass. As shown in Figure 2, we observe that FlashAttention achieves significant memory usage reduction for softmax attention approximation and even consumes much less memory than all considered approximate baselines under all sequence lengths. In terms of runtime, we notice that FlashAttention runs faster than most attention baselines under sequence lengths less than 2048 despite scaling quadratically, but EVA, along with other more efficient approximate variants, begin to catch up at longer sequence lengths. This implies that the quadratic computational costs of softmax attention still bottleneck its runtime performance, aligning with one of the main findings in Dao et al. (2022). According to this empirical study, we observe that FlashAttention offers a general and effective technique to speed up softmax attention; since many approximate variants (including EVA) exhibit a similar formulation to softmax attention (e.g., Equation 16), we expect they can also benefit from the optimized online softmax calculation technique and memory accesses of FlashAttention (Dao et al., 2022).

# F    EXPERIMENTS ON LONG RANGE ARENA

Long Range Arena (LRA; Tay et al., 2021) is a lightweight benchmark that assesses the ability of efficient attention methods to model long sequences in diverse domains. We follow the same hyper-parameter setup as Xiong et al. (2021b) to re-evaluate all attention baselines and report the

Table 15: Classification accuracy (%) on LRA benchmark with different efficient attention mechanisms.

| Model | ListOps | Text | Retrieval | Image | Pathfinder | Avg. |
|---|---|---|---|---|---|---|
| Softmax | **38.66** | 64.91 | 80.70 | 40.61 | 68.29 | 58.63 |
| Linformer | 38.21 | 53.91 | 77.66 | 39.40 | 66.44 | 55.12 |
| Performer | 29.84 | **65.30** | 77.70 | 38.29 | 66.39 | 55.50 |
| Reformer | 27.12 | 63.90 | 78.08 | 42.40 | 51.90 | 52.69 |
| Scatterbrain | 38.21 | 64.04 | 77.83 | 42.51 | 60.62 | 56.64 |
| Combiner | 38.26 | 63.98 | 81.47 | 42.80 | 55.94 | 56.49 |
| LARA | 37.10 | 64.62 | 80.82 | 38.99 | 68.96 | 58.10 |
| Nyströmformer | 38.46 | **65.28** | 80.44 | 39.71 | 68.98 | 58.57 |
| Local | 38.46 | 63.70 | 80.71 | 42.25 | 68.46 | 58.72 |
| Long-short | 38.56 | 63.46 | **81.73** | 40.54 | **71.28** | 59.11 |
| EVA | **38.61** | 64.31 | 80.21 | **43.24** | 70.90 | **59.45** |

comparison in Table 15. We observe that EVA largely improves previous RFA methods such as Performer (Choromanski et al., 2021) and LARA (Zheng et al., 2022b), and performs competitively with full softmax attention. Notably, EVA even achieves better average results over all tasks, with higher accuracy on Image and Pathfinder benchmarks, suggesting its capability of capturing long-term dependencies. For LRA benchmark, we set all attention-specific hyper-parameters to 128 (e.g., the number of landmarks in Nyströmformer (Xiong et al., 2021b) and LARA (Zheng et al., 2022b), the window size in local attention and Combiner (Ren et al., 2021), etc.). We set $|E| = 128$ and $C = 64$ by default for EVA without any further tuning and find this setup works well.

## G  CONNECTIONS TO OTHER ATTENTION MECHANISMS

### G.1  RFA, SOFTMAX ATTENTION, AND EVA

As mentioned in our main text, one of the main contributions of this work is to develop a more general framework that bridges RFA and conventional softmax attention. To see how EVA (Equation 13) achieves this goal formally, note that if either $|E| = M$ or $C = M$, EVA would be equivalent to standard softmax attention; while if we set $|E| = 0$ and $C = 1$, EVA would recover vanilla RFA.

### G.2  CONNECTIONS TO LARA

Notably, EVA and LARA (Zheng et al., 2022b) are two efficient attention mechanisms that are both built upon the self-normalized importance sampling (SNIS) formulation of RFAs. LARA (Zheng et al., 2022b) puts the main focus on the proposal distribution used in SNIS and tries to design importance sampling proposals that are closer to the true underlying distribution. The proposed usage of multiple proposals further improves the estimation quality of SNIS and achieves strong empirical performance while still keeping linear complexity.

In contrast to LARA, in this work we do not focus on the design choice of proposals used in importance sampling but aim to generalize the SNIS formulation further via control variates. As demonstrated in §3.2, our theory clearly delineates how the gap between such SNIS estimation and softmax attention can be closed by manipulating control variates. Since LARA and RFA are both SNIS estimators (their main difference lies in the choice of proposal distributions), our generalization also applies to LARA. To summarize, compared with LARA, EVA is a more general framework and improves conventional RFA from an orthogonal perspective.

### G.3  CONNECTIONS TO CLUSTERED ATTENTION

Clustered attention (Vyas et al., 2020) is an efficient attention mechanism that first clusters the set of queries into multiple groups, computes the mean centroid of each group, and then performs attention between query centroids and original key-value pairs. This framework is fast and effective and enjoys well-bounded approximation error.

Clustered attention and EVA share some similarities in two aspects. First, both of them adopt the partitioning technique to reduce the computational complexity while remaining effective; and secondly, both observe that the efficient attention mechanism can be improved by refining the approximation over specific elements. For instance, clustered attention can be improved (Vyas et al., 2020) by selecting top-$k$ key-value pairs that are most relevant to each centroid and then refining the approximation by recomputing attention weights over these keys using original queries; while EVA notices that we can directly employ the optimal control variate coefficient for a subset of key-value pairs ($m \in E$) while still remaining efficient, which yields a more accurate approximation.

Nevertheless, our main technical contribution is to develop a control variate formulation in the context of RFA and demonstrate that how RFA can be further improved locally. On the other hand, while clustered attention (Vyas et al., 2020) clusters queries, EVA partitions key-value pairs. This property makes EVA more amenable to the case of autoregressive language modeling since we do not impose clustering structures over the query set, and thus the causal relation among queries can be well maintained.

### G.4 Connections to Combiner

Combiner (Ren et al., 2021) is a recently proposed attention mechanism that also partitions the sequence into chunks combined with local attention. The key difference between EVA and Combiner is the motivation, where Combiner introduces a structured factorization over the attention probability distribution, while our approach is built from the control variate perspective.

### G.5 Connections to Scatterbrain

In this section, we show that Scatterbrain (Chen et al., 2021a) can be cast as a special case of our framework EVA, although they are proposed based on quite different motivations.

**A Brief Review of Scatterbrain.** Scatterbrain (Chen et al., 2021a) notes that sparse attention and RFA can approximate sharp and flat regions of the softmax attention matrix well, respectively. Based on this insight, Scatterbrain is proposed to first compute a Performer approximation to softmax attention and then cancel out the approximation error on critical regions via a sparse mechanism.

Specifically, Scatterbrain (Chen et al., 2021a) defines a sparse matrix $\mathbf{S} \in \mathbb{R}^{N \times M}$) so that for each $(n, m) \in \mathbf{S}$ that indexes a non-zero entry. For notational simplicity, we also denote $\mathrm{Supp}(\mathbf{S}) = \{(i, j)|S_{ij} \neq 0\}$ and $\mathrm{Supp}_n(\mathbf{S}) = \{m|S_{nm} \neq 0\}$. With random features $\phi(\cdot, \cdot)$ defined in Appendix A, we let

$$S_{nm} = \exp\left(\mathbf{q}_n^\top \mathbf{k}_m\right) - \phi(\mathbf{q}_n, \boldsymbol{\omega})^\top \phi(\mathbf{k}_m, \boldsymbol{\omega}).$$

We then add it back to the approximate output:

$$
\begin{aligned}
y_n' &= \sum_{m=1}^M \phi(\mathbf{q}_n, \boldsymbol{\omega})^\top \phi(\mathbf{k}_m, \boldsymbol{\omega}) \mathbf{v}_m + \mathbf{S}\mathbf{V} \\
&= \sum_{m=1}^M \phi(\mathbf{q}_n, \boldsymbol{\omega})^\top \phi(\mathbf{k}_m, \boldsymbol{\omega}) \mathbf{v}_m + \sum_{m' \in \mathrm{Supp}_n(\mathbf{S})} S_{nm'} \mathbf{v}_{m'} \\
&= \sum_{m \notin \mathrm{Supp}_n(\mathbf{S})} \phi(\mathbf{q}_n, \boldsymbol{\omega})^\top \phi(\mathbf{k}_m, \boldsymbol{\omega}) \mathbf{v}_m + \sum_{m' \in \mathrm{Supp}_n(\mathbf{S})} \exp\left(\mathbf{q}_n^\top \mathbf{k}_{m'}\right) \mathbf{v}_{m'}. \quad (29)
\end{aligned}
$$

The sparse mechanism can be thought of as modeling the error due to RFA and eliminating it on the support of $\mathbf{S}$. After the correction step, Scatterbrain further adds a post-hoc normalization step to obtain a normalized attention output:

$$
y_n = \frac{\sum_{m \notin \mathrm{Supp}_n(\mathbf{S})} \phi(\mathbf{q}_n, \boldsymbol{\omega})^\top \phi(\mathbf{k}_m, \boldsymbol{\omega}) \mathbf{v}_m + \sum_{m' \in \mathrm{Supp}_n(\mathbf{S})} \exp\left(\mathbf{q}_n^\top \mathbf{k}_{m'}\right) \mathbf{v}_{m'}}{\sum_{m \notin \mathrm{Supp}_n(\mathbf{S})} \phi(\mathbf{q}_n, \boldsymbol{\omega})^\top \phi(\mathbf{k}_m, \boldsymbol{\omega}) + \sum_{m' \in \mathrm{Supp}_n(\mathbf{S})} \exp\left(\mathbf{q}_n^\top \mathbf{k}_{m'}\right)}. \quad (30)
$$

Intuitively, Scatterbrain (Chen et al., 2021a) produces accurate approximation in the support of the sparse matrix and remains the random feature approximation outside the support.

**Scatterbrain is a Special Case of EVA.** For notational convenience, we denote $E := \text{Supp}_n(\mathbf{S})$. According to Proposition 1, suppose we employ optimal coefficients $\widehat{\boldsymbol{\beta}}_m$ for all entries in $\text{Supp}_n(\mathbf{S})$, and use the same coefficient $\widehat{\boldsymbol{\beta}}$ for all the remaining entries (in other words, we let $C = 1$ and the whole index set is only partitioned into two subsets $\{E, [M] \setminus E\}$). Then we have

$$\widetilde{g}_m(\boldsymbol{\omega}) = \begin{cases} g_m(\boldsymbol{\omega}) - \widehat{\boldsymbol{\beta}}_m h_m(\boldsymbol{\omega}) + \widehat{\boldsymbol{\beta}}_m \frac{\exp(\mathbf{q}_n^\top \mathbf{k}_m)}{Z} = \frac{\exp(\mathbf{q}_n^\top \mathbf{k}_m)\mathbf{v}_m}{Z}, & \text{if } m \in E, \\ g_m(\boldsymbol{\omega}) - \widehat{\boldsymbol{\beta}} h_m(\boldsymbol{\omega}) + \widehat{\boldsymbol{\beta}} \frac{\exp(\mathbf{q}_n^\top \mathbf{k}_m)}{Z}, & \text{if } m \notin E. \end{cases}$$

And the resulting estimator overall becomes

$$\begin{aligned}
\widetilde{g}(\boldsymbol{\omega}) &= \sum_{m=1}^{M} \widetilde{g}_m(\boldsymbol{\omega}) \\
&= \sum_{m \in E} \widetilde{g}_m(\boldsymbol{\omega}) + \sum_{m \notin E} \widetilde{g}_m(\boldsymbol{\omega}) \\
&= \sum_{m \in E} \frac{\exp(\mathbf{q}_n^\top \mathbf{k}_m)\mathbf{v}_m}{Z} + \sum_{m \notin E} \left( g_m(\boldsymbol{\omega}) - \widehat{\boldsymbol{\beta}} h_m(\boldsymbol{\omega}) + \widehat{\boldsymbol{\beta}} \frac{\exp(\mathbf{q}_n^\top \mathbf{k}_m)}{Z} \right) \\
&= \sum_{m \in E} \frac{\exp(\mathbf{q}_n^\top \mathbf{k}_m)\mathbf{v}_m}{Z} + \sum_{m \notin E} \left( g_m(\boldsymbol{\omega}) - \widehat{\boldsymbol{\beta}} h_m(\boldsymbol{\omega}) \right) + \widehat{\boldsymbol{\beta}} \sum_{m \notin E} \frac{\exp(\mathbf{q}_n^\top \mathbf{k}_m)}{Z} \\
&= \sum_{m \in E} \frac{\exp(\mathbf{q}_n^\top \mathbf{k}_m)\mathbf{v}_m}{Z} + \sum_{m \notin E} \left( g_m(\boldsymbol{\omega}) - \widehat{\boldsymbol{\beta}} h_m(\boldsymbol{\omega}) \right) + \widehat{\boldsymbol{\beta}} \left( 1 - \sum_{m \in E} \frac{\exp(\mathbf{q}_n^\top \mathbf{k}_m)}{Z} \right).
\end{aligned}$$

Scatterbrain (Chen et al., 2021a) can be a special case of this estimation algorithm if we set the proposal distribution to $q(\omega) = \mathcal{N}(\omega; 0, \mathbf{I})$, and estimate the normalizing constant as follows.

$$\begin{aligned}
Z &= \mathbb{E}_{\omega \sim q(\omega)} \left[ \frac{\mathcal{N}(\omega; 0, \mathbf{I}) \left( \sum_{m \in E} \xi(\mathbf{q}_n, \omega)^\top \xi(\mathbf{k}_m, \omega) + \sum_{m \notin E} \xi(\mathbf{q}_n, \omega)^\top \xi(\mathbf{k}_m, \omega) \right)}{q(\omega)} \right] \\
&= \sum_{m \in E} \exp(\mathbf{q}_n^\top \mathbf{k}_m) + \mathbb{E}_{\omega \sim q(\omega)} \left[ \frac{\mathcal{N}(\omega; 0, \mathbf{I}) \sum_{m \notin E} \xi(\mathbf{q}_n, \omega)^\top \xi(\mathbf{k}_m, \omega)}{q(\omega)} \right] \\
&\approx \sum_{m \in E} \exp(\mathbf{q}_n^\top \mathbf{k}_m) + \frac{1}{S} \sum_{s=1}^{S} \frac{\mathcal{N}(\omega; 0, \mathbf{I}) \sum_{m \notin E} \xi(\mathbf{q}_n, \omega)^\top \xi(\mathbf{k}_m, \omega)}{q(\omega_s)} \\
&= \sum_{m \in E} \exp(\mathbf{q}_n^\top \mathbf{k}_m) + \frac{1}{S} \sum_{s=1}^{S} \sum_{m \notin E} \xi(\mathbf{q}_n, \omega)^\top \xi(\mathbf{k}_m, \omega) \\
&= \sum_{m \in E} \exp(\mathbf{q}_n^\top \mathbf{k}_m) + \sum_{m \notin E} \phi(\mathbf{q}_n, \boldsymbol{\omega})^\top \phi(\mathbf{k}_m, \boldsymbol{\omega}) \\
&:= \sum_{m \in E} \exp(\mathbf{q}_n^\top \mathbf{k}_m) + \sum_{m \notin E} \widetilde{h}_m(\boldsymbol{\omega}),
\end{aligned}$$

where we define $\widetilde{h}_m(\boldsymbol{\omega}) = Z h_m(\boldsymbol{\omega})$, as in this case

$$g(\boldsymbol{\omega}) = \frac{1}{S} \sum_{s=1}^{S} \frac{p_n(\omega_s)}{q(\omega_s)} f(\omega_s) = \frac{1}{S} \sum_{s=1}^{S} \frac{1}{Z} \sum_{m=1}^{M} \xi(\mathbf{q}_n, \omega_s) \xi(\mathbf{k}_m, \omega_s) \mathbf{v}_m,$$

$$h(\boldsymbol{\omega}) = \frac{1}{S} \sum_{s=1}^{S} \frac{p_n(\omega_s)}{q(\omega_s)} = \frac{1}{S} \sum_{s=1}^{S} \frac{1}{Z} \sum_{m=1}^{M} \xi(\mathbf{q}_n, \omega_s) \xi(\mathbf{k}_m, \omega_s).$$

With these specifications, we obtain

$$
\begin{aligned}
\widetilde{g}(\boldsymbol{\omega}) &= \sum_{m \in E} \frac{\exp(\mathbf{q}_n^\top \mathbf{k}_m)\mathbf{v}_m}{Z} + \sum_{m \notin E}\left(g_m(\boldsymbol{\omega}) - \widehat{\boldsymbol{\beta}} h_m(\boldsymbol{\omega})\right) + \widehat{\boldsymbol{\beta}}\left(1 - \sum_{m \in E}\frac{\exp(\mathbf{q}_n^\top \mathbf{k}_m)}{Z}\right) \\
&= \sum_{m \in E} \frac{\exp(\mathbf{q}_n^\top \mathbf{k}_m)\mathbf{v}_m}{Z} + \sum_{m \notin E}\left(g_m(\boldsymbol{\omega}) - \widehat{\boldsymbol{\beta}} h_m(\boldsymbol{\omega})\right) + \widehat{\boldsymbol{\beta}}\frac{Z - \sum_{m \in E}\exp(\mathbf{q}_n^\top \mathbf{k}_m)}{Z} \\
&\approx \sum_{m \in E} \frac{\exp(\mathbf{q}_n^\top \mathbf{k}_m)\mathbf{v}_m}{Z} + \sum_{m \notin E}\left(g_m(\boldsymbol{\omega}) - \widehat{\boldsymbol{\beta}} h_m(\boldsymbol{\omega})\right) + \widehat{\boldsymbol{\beta}}\frac{\sum_{m \notin E}\widetilde{h}_m(\boldsymbol{\omega})}{Z} \\
&= \sum_{m \in E} \frac{\exp(\mathbf{q}_n^\top \mathbf{k}_m)\mathbf{v}_m}{Z} + \sum_{m \notin E}\left(g_m(\boldsymbol{\omega}) - \widehat{\boldsymbol{\beta}} h_m(\boldsymbol{\omega})\right) + \widehat{\boldsymbol{\beta}}\sum_{m \notin E} h_m(\boldsymbol{\omega}) \\
&= \frac{\sum_{m \in E}\exp(\mathbf{q}_n^\top \mathbf{k}_m)\mathbf{v}_m}{Z} + \sum_{m \notin E} g_m(\boldsymbol{\omega}) \\
&= \frac{\sum_{m \in E}\exp(\mathbf{q}_n^\top \mathbf{k}_m)\mathbf{v}_m}{Z} + \sum_{m \notin E} \frac{\frac{1}{S}\sum_{s=1}^{S}\xi(\mathbf{q}_n,\omega_s)\xi(\mathbf{k}_m,\omega_s)\mathbf{v}_m}{Z} \\
&= \frac{\sum_{m \in E}\exp(\mathbf{q}_n^\top \mathbf{k}_m)\mathbf{v}_m}{Z} + \sum_{m \notin E} \frac{\phi(\mathbf{q}_n,\boldsymbol{\omega})^\top \phi(\mathbf{k}_m,\boldsymbol{\omega})\mathbf{v}_m}{Z} \\
&\approx \frac{\sum_{m \notin E}\phi(\mathbf{q}_n,\boldsymbol{\omega})^\top \phi(\mathbf{k}_m,\boldsymbol{\omega})\mathbf{v}_m + \sum_{m' \in E}\exp\left(\mathbf{q}_n^\top \mathbf{k}_{m'}\right)\mathbf{v}_{m'}}{\sum_{m \notin E}\phi(\mathbf{q}_n,\boldsymbol{\omega})^\top \phi(\mathbf{k}_m,\boldsymbol{\omega}) + \sum_{m' \in E}\exp\left(\mathbf{q}_n^\top \mathbf{k}_{m'}\right)}
\end{aligned}
\tag{31}
$$

which is equivalent to Scatterbrain (Equation 30). Note that this equivalence would hold irrespective of the choice of shared coefficients $\widehat{\boldsymbol{\beta}}$, which possibly indicates that the formulation of Scatterbrain limits the potential benefit of optimizing control variates under our framework.

