# OpenReview forum: "Efficient Attention via Control Variates"
_ICLR.cc/2023/Conference — ICLR 2023 notable top 5%_

### Official Review · Reviewer_xJRA · 2022-10-22

**Confidence:** 4
**Correctness:** 3
**Technical Novelty And Significance:** 2
**Empirical Novelty And Significance:** 2
**Recommendation:** 8

**Clarity, Quality, Novelty And Reproducibility:**

Clarity: The paper is well-written.

Quality: Overall good, the evaluation methodology is standard and complete (thank you for evaluating runtime instead of just FLOPs). A few concerns:

1. There are gaps between EVA and standard attention in language modeling on Books3 and Wikitext-103. It appears the gap on Books3 can be closed with a larger model, but the gap on WikiText3 has not been cloased (even when reducing the parameters of the attention model). This does not bode well for scaling to larger models -- can you try to close the gap more or explain why a gap remains?

2. The results on efficiency are very exciting! It would be stronger to also compare against some recent "memory-efficient attention" mechanisms, such as those in xformers or FlashAttention. I believe those methods will also have memory that scales linearly (but compute scaling quadratically).

Novelty: The contribution is a little bit incremental, but I think it provides some nice insight and recontextualization of existing methods.

Reproducibility: Seems clear from the paper.

**Strength And Weaknesses:**

+ A nice analysis of RFA that provides some insight into what the method is doing.
+ Interesting method building on the analysis.

- Some weak empirical results suggest that this method may not scale to larger models (this is common for RFA-based efficient attention mechanisms).
- A couple missing baselines.

**Summary Of The Paper:**

This paper studies the gap between random-feature-based attention (RFA) and standard attention. The paper characterizes the gap using control variates and proposes a novel efficient attention mechanism based on the analysis. The authors evaluate their method EVA in terms of model quality, efficiency, and runtime.

**Summary Of The Review:**

Overall I think this paper makes a positive contribution to the literature. The results also show minor improvements over previous efficient attention mechanisms. A few gaps and baselines remain, but I think this paper is above the acceptance threshold.

Update after rebuttal: my concerns have been addressed, so I am increasing my score to an 8.

---

> ### Author Response · Authors · 2022-11-17
> **Response to Reviewer xJRA**
>
> Thanks for your feedback and thoughtful comments! Here are our responses:
> > **Q1**: …but the gap on WikiText3 has not been closed (even when reducing the parameters of the attention model). This does not bode well for scaling to larger models -- can you try to close the gap more or explain why a gap remains?
>
> **A1**: Thanks for the suggestion! In the Wikitext-103 benchmark, we spot an undesirable behavior of EVA that disables attention dropout during training, while standard softmax attention in FairSeq does with dropout rate of 0.1 for proper regularization. Enabling such attention dropout (with the same rate of 0.1 in Equation 16) effectively prevents EVA from overfitting during training, resulting in the same perplexity as exact softmax attention (see the comparison below and Table 5 in the updated manuscript). When further doubling the number of Transformer layers (from 16 to 32, as in previous work [1]), we observed that EVA **scales just as effectively as softmax attention in achieving the same perplexity score**. This validates the effectiveness of the finer-grained control variate formulation of EVA, which enlarges its approximation capacity.
>
> (Val. = Validation set, PPL = perplexity, CW = Context Window size)
> | Model | # Params. | Val. PPL w/ CW 256| Test PPL w/ CW 256 | Val. PPL w/ CW 480| Test PPL w/ CW 480|
> | --- | --- |  --- |  --- |  --- |  --- |
> | EVA (w/o attention dropout) | 247M | 19.6 | 20.2 | 19.2 | 19.9 |
> | EVA (w/ attention dropout) **(New)**  | 247M | 18.8 | 19.4 | 18.5 | 19.1 |
> | Softmax | 247M | 18.8 | 19.5 | 18.4 | 19.1 |
> | EVA  (w/ attention dropout) **(New)** | 450M | 17.9 | 18.6 | 17.7 | 18.3 |
> | Softmax | 449M | 17.9 | 18.5 | - | - |
>
> > **Q2**: The results on efficiency are very exciting! It would be stronger to also compare against some recent "memory-efficient attention" mechanisms, such as those in xformers or FlashAttention. I believe those methods will also have memory that scales linearly (but compute scaling quadratically).
>
> **A2**: We have added a new paragraph (Appendix E.4) as well as an empirical study to compare EVA and memory-efficient attention. Here are our main findings:
>
> 1. We compared efficient attention variants (including EVA) against FlashAttention [2], one of the memory-efficient attention mechanisms (the xFormers codebase implemented several variants of mem-efficient attention; however, we find these variants (including the CUTLASS-based implementation) consumes almost the same amount of memory as FlashAttention but runs much slower. Therefore, we mainly consider FlashAttention for the comparison).
> 2. As shown in Figure 2, mem-efficient attention does an excellent job in scaling down the memory usage, which enables exact softmax attention to be computed in much less memory usage than EVA (along with other approximate methods) even under much longer sequences (e.g., 8192). The run-time speedup is also significant (often more than 2x faster than vanilla softmax attention); however, since the computation involved in FlashAttention still scales quadratically, these linear approximate variants (including EVA) begin to catch up under longer sequence lengths. The result indicates that the quadratically many computations in softmax attention still bottlenecks its performance, even with optimized memory accesses. This empirical finding is also in line with [2].
> 3. We also observe that FlashAttention, along with the family of memory-efficient attentions, offer very general techniques to accelerate softmax attention-like computations. Since many approximate attention methods (e.g., EVA, Linformer, Local attention, Long-short, and many others considered in this work) exhibit a similar formulation to softmax attention (i.e., perform softmax-style normalization and weighted average over value vectors), they are also expected to benefit from the online softmax calculation and IO-level optimization (despite the significant engineering effort). This is an exciting future research direction, as it is a modular approach to further accelerating general attention computations.
>
> [1] Kasai, Jungo, et al. "Finetuning Pretrained Transformers into RNNs." Proceedings of the 2021 Conference on Empirical Methods in Natural Language Processing. 2021.
>
> [2] Dao, Tri, et al. "FlashAttention: Fast and Memory-Efficient Exact Attention with IO-Awareness." arXiv preprint arXiv:2205.14135 (2022).

---

> > ### Comment · Reviewer_xJRA · 2022-11-17
> > **Response**
> >
> > Thank you for your response -- the results on scaling look very promising, and the experiments address my concerns. I am upgrading my score to reflect this.

---

> > > ### Author Response · Authors · 2022-11-18
> > > **Response to Reviewer xJRA**
> > >
> > > We are glad our response addressed your concerns. Thank you very much for your insightful feedback and kind reply!

---

### Official Review · Reviewer_KXnA · 2022-10-24

**Confidence:** 3
**Correctness:** 4
**Technical Novelty And Significance:** 4
**Empirical Novelty And Significance:** 3
**Recommendation:** 8

**Clarity, Quality, Novelty And Reproducibility:**

The paper is generally quite clear apart from the weaknesses pointed out above.

The quality of the theoretical and experimental results is extremely high.

In this reviewer's opinion, the novelty is high. Connecting the attention mechanism with known results on control variates and proposing practically efficient decompositions is non-trivial.
The authors have provided an interesting derivation of ScatterBrain as a special case of their control variate approach in Appendix G.

The github link to the code is apparently not shared and is hence an impediment to reproducibility.

**Strength And Weaknesses:**

Strengths

1.
The proposed EVA approach is motivated quite well and the paper is quite clear.

2.
The theoretical results (Propositions 1, 2 and ) in the paper are impressive and appear to be quite sound.
(I didn't check the extensive appendix though.)

3.
The experimental validation is quite convincing on a variety of problems.

Weaknesses

1.
Certain choices are not well-motivated. For example, rather than learning the partitioning scheme via a K-means-like algorithm for the keys receiving clustered coefficients (instead of optimized control coefficients), the clusters (partitions) are chosen as contiguous chunks. This is called out in the final section though.

2.
It's not clear how hyper-parameters such as E and C in EVA were tuned.

**Summary Of The Paper:**

Firstly, as in Zheng 2022, the authors interpret the computationally efficient RFA (randomized feature-based attention) as a SNIPS estimate (self-normalized importance sampling estimate). They then use Vlassis' control variate formulation of SNIPS to link exact attention with RFA via a single global control covariate coefficient. This naturally suggests using more control coefficients to obtain more accurate coefficients while maintaining efficiency in the form of exact coefficient estimates for certain keys most correlated with the query and coefficients per cluster for the remaining keys. This new approach is termed EVA and empirical demonstrations of its efficacy are quite convincing. On ImageNet using the DeiT-Tiny-784 architecture, EVA outperforms Zheng's LARA approach and other forms of efficient attention in Table 2, while in Table 1, using a different architecture, EVA outperforms all efficient attention variants, but not the slower exact attention mechanism itself. In 3 NLP applications (MLM, MT, Autoregressive LM), EVA again outperforms other efficient variants in terms of BLEU score or test perplexity, although there were issues with replicating Kasai's perplexity for exact attention.

**Summary Of The Review:**

I strongly recommend acceptance on account of the novel theoretical results and the extensive convincing experimental results.
It should be straightforward to address the minor concerns regarding clarity and reproducibility noted above.

---

> ### Author Response · Authors · 2022-11-17
> **Response to Reviewer KXnA**
>
> Thanks for your feedback and thoughtful comments! Here are our responses:
> > **Q1**: Certain choices are not well-motivated. For example, rather than learning the partitioning scheme via a K-means-like algorithm for the keys receiving clustered coefficients (instead of optimized control coefficients), the clusters (partitions) are chosen as contiguous chunks. This is called out in the final section though.
>
> **A1**: Thanks for your comment! We agree that learning a partition scheme is a more general way to define our subproblems of interest. Nevertheless, since our main contribution in this work is proposing a decomposed representation for random-feature-based approximate attention, we only adopt the most straightforward partitioning strategy (i.e., contiguous chunking) to explore the benefit of our control variate framework. As mentioned in the final section, it would be interesting to investigate more sophisticated partitioning strategies, which can be either guided by task-specific biases (e.g., syntactic parses for language tasks or other hierarchical tree structures) or learnable (e.g., similar to Routing Transformer [7]). However, we feel this could extend our framework in a non-trivial way and fall beyond the scope of this work. In particular, it might also introduce other challenges (such as how to condition the partitioning on the task information or deal with the learnable clustering centroids, etc.) and inspire better formulations of the introduced control variates (e.g., eliminating the need to approximate the exact computation).
>
> > **Q2**: It's not clear how hyper-parameters such as E and C in EVA were tuned.
>
> **A2**: We provide more details for the choice of $|E|$ and $C$ in the updated manuscript (Appendix E). In general, we follow previous practices to choose $C$ and $|E|$ for each task specifically so that the overall computation of EVA remains comparable to previous baselines.
>
> 1. For image classification, we set both $C$ and $|E|$ to 49 according to [1,2] regardless of the model architecture (PVT/DeiT) and actual sequence length (the sequence length could vary among {49, 196, 784, 3136} in PVT while {196, 784} in DeiT).
> 2. For language tasks, denote $N$ as the sequence length for each task, $C = N / \lambda$ and $|E| = kC$. In general, we select $\lambda \in \{4,8,16\}$ based on **previous practices** and select $k \in \{0.5,1,2,4\}$ based on the **validation performance**. We find that across various tasks, $k = 2$ attains the best tradeoff between performance and compute; as a result, $k$ is set to 2 by default for all tasks unless specified otherwise.
>    1. For masked language modeling on Books3 corpus, we set $\lambda$ to 16 by default (resulting in $C = 128$) as in [3];
>    2. For autoregressive language modeling on Wikitext-103, we set $\lambda$ to 8 ($C = 64$) similar to the setup in [5,6];
>    3. For machine translation, we let $C = 8$ (approximately amounts to $\lambda = 4$; we set $C$ directly since this task involves variable-length sentence pairs) due to shorter sequences [1,4].
> 3. For the ablation study, we vary the value of $C$ and $|E|$ to test their effect on the performance.
>
> > **Q3**: The github link to the code is apparently not shared and is hence an impediment to reproducibility.
>
> **A3**: Thanks for your feedback! We will make our code public upon acceptance.
>
> [1] Zheng, Lin, Chong Wang, and Lingpeng Kong. "Linear Complexity Randomized Self-attention Mechanism." International Conference on Machine Learning. PMLR, 2022.
>
> [2] Wang, Wenhai, et al. "Pvt v2: Improved baselines with pyramid vision transformer." Computational Visual Media 8.3 (2022): 415-424.
>
> [3] Xiong, Wenhan, et al. "Simple Local Attentions Remain Competitive for Long-Context Tasks." Proceedings of the 2022 Conference of the North American Chapter of the Association for Computational Linguistics: Human Language Technologies. Association for Computational Linguistics, 2022.
>
> [4] Ma, Xuezhe, et al. "Luna: Linear unified nested attention." Advances in Neural Information Processing Systems 34 (2021): 2441-2453.
>
> [5] Peng, Hao, et al. "Random Feature Attention." International Conference on Learning Representations. 2021.
>
> [6] Schlag, Imanol, Kazuki Irie, and Jürgen Schmidhuber. "Linear transformers are secretly fast weight programmers." International Conference on Machine Learning. PMLR, 2021.
>
> [7] Roy, Aurko, et al. "Efficient content-based sparse attention with routing transformers." Transactions of the Association for Computational Linguistics 9 (2021): 53-68.

---

> > ### Comment · Reviewer_KXnA · 2022-11-21
> > **Response**
> >
> > I wish to thank the authors for satisfactorily addressing my concerns !
> > (My high score remains unchanged.)

---

> > > ### Author Response · Authors · 2022-11-22
> > > **Response to Reviewer KXnA**
> > >
> > > We are glad our response addressed your concerns. Thank you very much for your insightful comment and kind reply!

---

### Official Review · Reviewer_B8ci · 2022-10-25

**Confidence:** 3
**Correctness:** 2
**Technical Novelty And Significance:** 3
**Empirical Novelty And Significance:** 2
**Recommendation:** 6

**Clarity, Quality, Novelty And Reproducibility:**

The reviewer agrees that the work tackles an essential problem. However, several parts need further clarification by the authors to conclude the contributions of the work.

**Strength And Weaknesses:**

Strength:

- The paper develops an efficient attention mechanism EVA via control variates. This novel attention mechanism is efficient and can bridge the gap between RFA and exact softmax attention. The EVA attains a good trade-off between modeling quality and efficiency.


Weakness

- The author should compare more state-of-the-art linear attention like Flowformer or FLASH.

**Summary Of The Paper:**

This work improves Random-feature-based attention through the lens of control variates. The paper develops a more flexible form of control variates, which forms a novel attention mechanism that significantly reduces the approximation gap while maintaining linear complexity. The EVA achieves competitive results on several tasks.

**Summary Of The Review:**

In this paper, the author proposes an efficient attention mechanism called EVA with the help of control variates.

---

> ### Author Response · Authors · 2022-11-17
> **Response to Reviewer B8ci**
>
>
> Thanks for your thoughtful comments! Here are our responses:
> > **Q**: The author should compare more state-of-the-art linear attention like Flowformer or FLASH.
>
> **A**: In our work, we compared EVA against LS and combiner, which are shown to be strong baselines across various tasks [2,3]. Besides, we also notice that both EVA and Flowformer [1] are evaluated on ImageNet1k classification under similar setups. The following tables list the comparison between them, where the numbers are taken directly from the paper. We observe that EVA performs better than Flowformer under comparable numbers of parameters and FLOPs. We attribute this to the significantly improved formulation in EVA, which processes sequences in a finer-grained manner thanks to the decomposed control variate representation.
>
> Image classification accuracy on ImageNet1k with DeiT-Small architecture
> | Model | # Params | FLOPs | Top-1 Acc. |
> |  ---- | :----: |  :----: | :----: |
> | Flowformer | 22M | 4.2G | 80.0 |
> | EVA | 22M | 4.4G | 80.65 |
>
> Image classification accuracy on ImageNet1k with Hierarchical ViT architecture
> | Model | # Params | FLOPs | Top-1 Acc. |
> |  ---- | ---- |  ---- |---- |
> | Flowformer | 41M | 6.3G | 80.6 |
> | EVA | 36M | 7.4G | 83.71 |
>
> To ensure a head-to-head comparison between EVA and its aforementioned baselines, including 1) Flowformer and 2) FLASH (a strong baseline for language modeling tasks, which combines linear and local attention along with Transformer architecture improvements [2]), we train all these baselines on the Wikitext-103 benchmark following the same hyper-param setting as in Flowformer.
>
> Language modeling performance on Wikitext-103 (6-layer Transformer)
> | Model | Perplexity |
> | --- | --- |
> | Flowformer | 30.8 |
> | EVA | **28.38** |
> | FLASH | 29.12   |
>
> In general, we find that EVA still outperforms these baselines despite FLASH being more competitive. We will conduct more evaluations and incorporate the results into the revision.
>
> [1] Wu, Haixu, et al. "Flowformer: Linearizing Transformers with Conservation Flows." International Conference on Machine Learning. PMLR, 2022.
>
> [2] Hua, Weizhe, et al. "Transformer quality in linear time." International Conference on Machine Learning. PMLR, 2022.
>
> [3] Zhang, Jun, et al. "CAB: Comprehensive Attention Benchmarking on Long Sequence Modeling." arXiv preprint arXiv:2210.07661 (2022).

---

### Official Review · Reviewer_rNYj · 2022-10-25

**Confidence:** 4
**Correctness:** 4
**Technical Novelty And Significance:** 3
**Empirical Novelty And Significance:** 3
**Recommendation:** 8

**Clarity, Quality, Novelty And Reproducibility:**

The EVA implementation includes the heuristics to approximate control variate computing. It might be not easy to reproduce author's results without code provided.

**Strength And Weaknesses:**

Strength
- The idea is easy to follow, and the paper is well-written.
- The mathematic analysis is given to explain the dissecting RFA with control variates.
- The EVA attention with simplification achieving higher accuracy than softmax in image classification is impressive.

Weakness

- Overall, I would say the paper is in good shape in terms of detailed mathematic analysis and solid experiments on multiple vision and natural language tasks. My main concern is about the usage in the practical. Though the EVA achieves higher accuracy (very subtle) than softmax in table 1, the softmax is still a better choice in terms of accuracy, #params, and FLOPs. According to numbers reported in all tables, softmax still dominates the performance and has comparable #param and FLOPs to EVA.
- The not-well-designed approximation in computing control variate could impact the EVA's performance. As no code is provided, it is hard to trace the author's implementation.

**Summary Of The Paper:**

In this work, the authors propose an efficient attention EVA, where the work reals that exact softmax attention can be recovered from RFA by manipulating each control variate. The mathematic analysis is provided to prove the dissecting RFA with control variates. And the authors also implemented their EVA and applied it to image classification and natural language tasks, showing performance boosting over SOTA.

**Summary Of The Review:**

Though in terms of accuracy/#param/Flops, the EVA might not beat SoftMax completely, the work with theoretical evidence for approximating the  SoftMax attention can provide insights into the future works of light-weight attention development. The paper is in well-written, and the experiments are solid. Hence, the work reaches the acceptance bar.

---

> ### Author Response · Authors · 2022-11-17
> **Response to Reviewer rNYj**
>
> Thanks for your feedback and thoughtful comments! Here are our responses:
> > **Q1**: My main concern is about the usage in the practical. Though the EVA achieves higher accuracy (very subtle) than softmax in table 1, the softmax is still a better choice in terms of accuracy, #params, and FLOPs. According to numbers reported in all tables, softmax still dominates the performance and has comparable #param and FLOPs to EVA.
>
> **A1**: Thanks for your comment. We agree that softmax attention is still a more effective method to process sequences than most of its efficient variants in general. However, its quadratic complexity limits its usage in the long-sequence setting (Figure 1(a) and 1(b)). This is the reason why we propose EVA to **approximate** softmax attention (e.g., see Proposition 1 & 2) with lower computational complexity. This point is also addressed in Appendix E.4, where we find that: for relatively shorter sequences, EVA often performs better than softmax attention under comparable running time (Table 1), possibly due to the formed structure of subsequences. While for longer sequences, EVA runs much faster while still achieving comparable performance (c.f. Figure 1(c), Table 5 and Table 7). This indicates that it is possible to improve conventional attention mechanisms further to eliminate its quadratic computation bottleneck without affecting performance.
>
> > **Q2**: The not-well-designed approximation in computing control variate could impact the EVA's performance. As no code is provided, it is hard to trace the author's implementation.
>
> **A2**: We agree that the approximation in computing the expectation of control variates is crude; however, we provide an informal justification in Appendix C for this approximation and find it works well across various tasks. We leave the in-depth investigation for more accurate approximation in future work. As for the implementation, we closely follow the listed pseudo-codes (Algorithm 1 & 2 in Appendix) to implement EVA in practice; our codebase will also be publicly available upon acceptance.

---

### Author Response · Authors · 2022-11-17
**Summary of Updates**

We thank the reviewers for their insightful suggestions and comments! We revised the paper accordingly, and here is a summary of updates:

1. In Section 5, we fix an undesirable behavior in EVA’s implementation, update the experimental results of Wikitext-103 autoregressive language modeling and find that EVA achieved almost the same perplexity as full softmax attention (Table 5).
2. In Section 6, we discuss the comparison between approximate attention methods and memory-efficient exact attention mechanisms; we also include an empirical study to compare the efficiency of these methods in Appendix E.
3. In Appendix E, we provide more details about how the hyper-parameters $E$ and $C$ in EVA are selected.
4. We fix the typo and grammar accordingly.

---

### Decision · Program_Chairs · 2023-01-20

**Decision:**

Accept: notable-top-5%

**Justification For Why Not Higher Score:**

N/A

**Justification For Why Not Lower Score:**

The work can have a large impact on the community given the high interest in transformers-like architectures at a lower cost.

**Metareview: Summary, Strengths And Weaknesses:**

The paper studies the gap between random-feature-based attention (RFA) and standard attention. As a result of this study, the authors characterize the gap using control variates and propose a novel efficient attention mechanism based on the analysis which retains the linear runtime and space complexity of RFA. The authors evaluate their method EVA in terms of model quality, efficiency, and runtime and obtain SOTA results on both vision and language tasks.

All reviewers and AC agree that the paper should be accepted as its impact can be significant on the computational/memory cost of transformers without compromising performance.

**Note From Pc:**

if the above contains the word "oral" or "spotlight" please see: "oral" presentation means -> notable-top-5% and "spotlight" means -> notable-top-25%. As stated in our emails, we are disassociating presentation type from AC recommendations